# CALM: Culturally Self-Aware Language Models

**Lingzhi Shen♦, Xiaohao Cai♦, Yunfei Long♣, Imran Razzak★, Guanming Chen♦, Shoaib Jameel♦**
♦University of Southampton, Southampton, United Kingdom
♣Queen Mary University of London, London, United Kingdom
★Mohamed bin Zayed University of Artificial Intelligence, Abu Dhabi, United Arab Emirates

## Abstract

Cultural awareness in language models is the capacity to understand and adapt to diverse cultural contexts. However, most existing approaches treat culture as static background knowledge, overlooking its dynamic and evolving nature. This limitation reduces their reliability in downstream tasks that demand genuine cultural sensitivity. In this work, we introduce CALM, a novel framework designed to endow language models with cultural self-awareness. CALM disentangles task semantics from explicit cultural concepts and latent cultural signals, shaping them into structured cultural clusters through contrastive learning. These clusters are then aligned via cross-attention to establish fine-grained interactions among related cultural features and are adaptively integrated through a Mixture-of-Experts mechanism along culture-specific dimensions. The resulting unified representation is fused with the model's original knowledge to construct a culturally grounded internal identity state, which is further enhanced through self-prompted reflective learning, enabling continual adaptation and self-correction. Extensive experiments conducted on multiple cross-cultural benchmark datasets demonstrate that CALM consistently outperforms state-of-the-art methods.

## 1 Introduction

As anthropologist Clifford Geertz stated in *The Interpretation of Cultures* [1], a foundational text in cultural anthropology, culture is not merely the observable pattern of behaviour but "a system of inherited conceptions expressed in symbolic forms by means of which people communicate, perpetuate, and develop their knowledge about and attitudes toward life." In language understanding, cultural awareness enables individuals to go beyond linguistic form [2] and comprehend how meaning is shaped by social and cultural context. It involves internalizing shared norms and situational conventions [3], allowing interpretation to consider not only literal meaning but also intent and appropriateness. Without such awareness, language understanding becomes superficial and contextually detached, often leading to pragmatic errors or misunderstanding.

In language models, cultural awareness is key to social sensitivity and contextual appropriateness [4], enabling model behaviour to align with diverse communicative norms and expectations [5, 6]. This capability is essential for a wide range of cross-cultural applications, including dialogue systems [7], educational technologies [8], and content moderation [9]. Grounding language in cultural understanding allows models to move beyond surface-level fluency and generate responses that are contextually coherent and socially responsible, thereby contributing to greater fairness and inclusivity in model outputs [10]. Figure 1 illustrates how an AI system lacking cultural grounding can misinterpret context and offer inappropriate suggestions.

Recent work has increasingly highlighted the importance of cultural awareness in large language models (LLMs). CultureBank [11] builds a large-scale knowledge base by extracting structured

---

*The source code is available at https://github.com/slz0925/CALM.

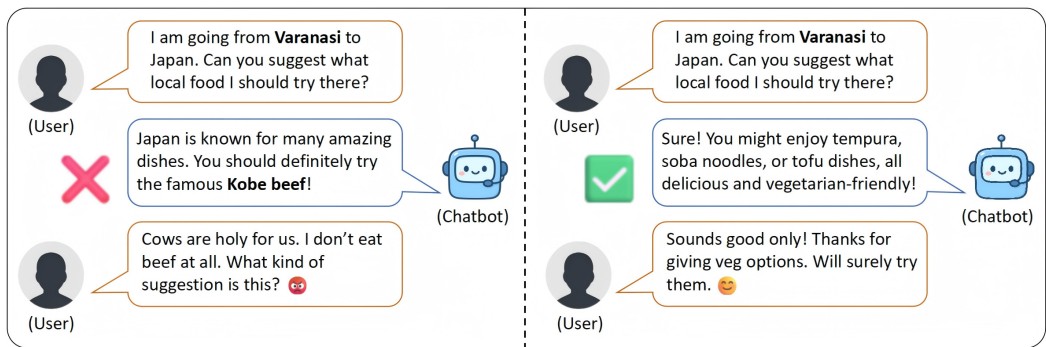

Figure 1: A case of culturally sensitive food recommendation. Left: AI assistant recommends beef to a user from Varanasi, resulting in discomfort. Right: Recommendations lead to a satisfying outcome.

cultural descriptors from online user-generated narratives, but its reliance on handcrafted heuristics treats culture as static labels and limits contextual adaptability. CulturePark [12] generates cross-cultural dialogue data through simulated multi-agent interactions, yet its synthetic nature lacks the grounding and diversity of real human communication. SeaLLMs [13] adopts a language-as-culture perspective by combining continual multilingual pretraining, vocabulary expansion, and self-preference alignment to better capture the sociopragmatic features of Southeast Asian languages. However, its focus on linguistic structures leaves higher-level cultural reasoning and transfer across non-linguistic dimensions underexplored.

In this paper, we introduce CALM, a framework for *Culturally self-Aware Language Models*. CALM first constructs an abstract cognitive space that extracts latent cultural signals and explicit cultural concepts beyond task semantics. These features are organized through within-type contrastive learning into structured semantic distributions that form clear cultural clusters. In the identity alignment pool, cross-attention [14] captures the interactions among related cultural features across these clusters, generating finely aligned cultural representations. The aligned representations are dynamically routed through a Mixture-of-Experts (MoE) [15] with an expert selection mechanism [16], which enables culturally informed specialization along communicative dimensions. Residual fusion preserves essential cultural signals from the original features and combines them with high-level reasoning from experts to form a unified cultural self-representation. Finally, CALM performs reflective reasoning through a self-corrective loop that integrates culturally conditioned prompt generation, culturally grounded reasoning, and identity calibration mechanism, allowing the model to self-adjust its cultural alignment when outputs deviate from its internal cultural representation.

**Key Contributions:** CALM first introduces a novel disentanglement mechanism that separates task semantics from both explicit and latent cultural features within an abstract cognitive space. Building on this foundation, a structured identity alignment pool unifies cultural signals through contrastive learning and cross-attention, producing coherent cultural representations. These representations are dynamically routed through a culture-informed MoE module that enables adaptive reasoning along communicative dimensions. Finally, CALM incorporates a self-corrective loop that ensures continual cultural adaptability. Together, these innovations advance the frontier of cultural reasoning, as demonstrated by superior performance across diverse benchmarks and supported by extensive ablation, qualitative, and quantitative analyses.

## 2   Related Work

Prior work on cultural alignment in language models falls into two main paradigms: **data-centric training** and **prompt-based inference**. Data-centric approaches embed culture into model parameters via pretraining or fine-tuning on curated resources such as multilingual corpora, annotated datasets, and knowledge bases [17–19]. These methods typically encode culture as facts, values, symbols, or demographic attributes [20–23], often assuming culture is static and fully specifiable during training. While these approaches can improve cultural factuality, they also risk reinforcing biases inherent in the training data [24] and often treat culture as a static construct, limiting adaptability to evolving cultural contexts [25]. In contrast, prompt-based inference techniques manipulate inputs at runtime to simulate cultural perspective-taking using anthropological framing, demographic priming, or national

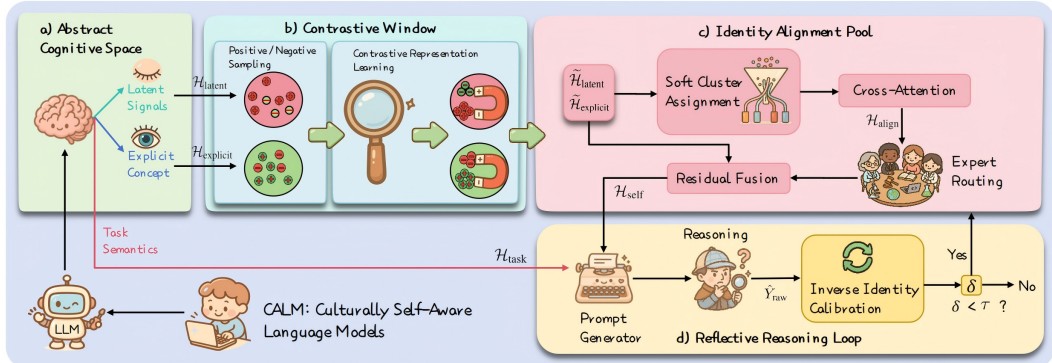

Figure 2: Overview of the CALM framework, comprising four stages that progressively embed cultural awareness into internal reasoning and enable self-consistency calibration.

---

**Algorithm 1** Pseudocode of CALM

---

**Require:** Inputs $\mathcal{D}$, parameters $\Theta$, epochs $N$, batch size $B$, learning rate $\eta$, and criterion $\tau$
**Ensure:** Task-appropriate, culturally aligned output $\hat{o}$
1: **for** $i = 1$ to $N$ **do**
2:     Sample batch $X \sim \mathcal{D}$ of size $B$
3:     $[\mathcal{S}, \mathcal{E}, \mathcal{L}] \leftarrow \textbf{Extract}_\Theta(X)$;   $[\widetilde{\mathcal{E}}, \widetilde{\mathcal{L}}] \leftarrow \textbf{Contrast}(\mathcal{E}, \mathcal{L})$
4:     $[\mathcal{C}_{\text{explicit}}, \mathcal{C}_{\text{latent}}] \leftarrow \textbf{Partition}(\widetilde{\mathcal{E}}, \widetilde{\mathcal{L}})$;   $\mathcal{A} \leftarrow \textbf{Interact}(\mathcal{C}_{\text{latent}}, \mathcal{C}_{\text{explicit}})$
5:     $\{\mathcal{H}_d\}_{d=1}^3 \leftarrow \textbf{SparseMap}(\mathcal{A})$;   $\mathcal{I} \leftarrow \textbf{Integrate}(\{\mathcal{H}_d\}, \widetilde{\mathcal{E}}, \widetilde{\mathcal{L}}, \mathcal{S})$
6:     $P \leftarrow \textbf{SelfPrompt}(\mathcal{S}, \mathcal{I})$;   $\hat{o} \leftarrow \textbf{OutputHead}(\mathcal{I}, P)$
7:     $\mathcal{I}' \leftarrow \textbf{IdentityEstimate}(\hat{o})$
8:     **if** $\text{sim}(\mathcal{I}, \mathcal{I}') < \tau$ **then**
9:         $\mathcal{I} \leftarrow \textbf{Reflect}(\mathcal{I}, \mathcal{I}')$;   $\hat{o} \leftarrow \textbf{OutputHead}(\mathcal{I}, P)$
10:    Update $\Theta$ using loss $\mathcal{L}(\hat{o})$
11: **return** $\hat{o}$

---

identity prompts [26–28]. These methods are flexible and training-free [29] but treat culture as an external constraint, lacking integration into the model's reasoning process and often yielding brittle or inconsistent behavior across different scenarios [30–32]. To move beyond this dichotomy, CALM models culture as an internal, dynamic reasoning state, enabling deeper and more adaptable cultural awareness.

## 3 Our Novel CALM Framework

As shown in Figure 2 and Algorithm 1, inspired by sociocultural theories of meaning construction [33] and metacognitive regulation [34], CALM models cultural awareness as a dynamic internal reasoning process. This process is implemented through a modular closed-loop architecture that consists of four components: perception, structural induction, identity construction, and reflective correction. Each component corresponds to a specific stage of cultural cognition.

### 3.1 Abstract Cognitive Space

Grounded in sociocultural theories of communication [35], we conceptualize culturally grounded language understanding as the interaction of three cognitive dimensions: task semantics, explicit cultural concepts, and latent cultural signals. This triadic decomposition reflects the layered structure of human communication, encompassing semantic content, symbolic representation, and pragmatic inference. To operationalize this, CALM employs a unified LLM backbone that disentangles inputs into three parallel representations, each aligned with one of these dimensions.

Given an input sequence $X = \{x_1, x_2, \ldots, x_n\}$, we represent it as:

$$\mathcal{H}_{\text{ACS}} = \mathcal{H}_{\text{task}} \oplus \mathcal{H}_{\text{explicit}} \oplus \mathcal{H}_{\text{latent}}, \tag{1}$$

where $\mathcal{H}_{\text{task}} = f_{\text{task}}(X)^1$, $\mathcal{H}_{\text{explicit}} = f_{\text{explicit}}(X)^2$, $\mathcal{H}_{\text{latent}} = f_{\text{latent}}(X)^3$, and $\oplus$ denotes the concatenation of feature streams derived from the same LLM encoder.

The resulting $\mathcal{H}_{\text{ACS}}$ in Eqn (1) encodes a multi-level abstraction that disentangles semantic intent, symbolic culture, and pragmatic norms, forming the cognitive foundation of CALM. While $\mathcal{H}_{\text{ACS}}$ is not used as a standalone variable in downstream equations, its subcomponents serve as the core representational inputs for all subsequent modules. Further theoretical explanations of the abstract cognitive space are elaborated in Appendix B.1.

## 3.2 Contrastive Window

To enhance the structural regularity and discriminability of the abstract cognitive space, CALM incorporates a contrastive window that refines explicit and latent cultural signals through type-specific contrastive learning. These two feature types originate from distinct linguistic levels: explicit concepts capture lexical and phrase-level elements (e.g., idioms, role titles) [40], while latent signals encode sentence- and discourse-level stylistic traits (e.g., tone, formality) [41]. Contrastive learning encourages culturally coherent subspaces by maximizing intra-cultural similarity and minimizing inter-cultural overlap. Specifically, we apply type-specific projection heads (separate for explicit and latent features) on LLM-encoded hidden states, obtaining normalized cultural embeddings via a SimCLR-style setup [42]. Positive pairs are sampled from semantically similar inputs within the same cultural group; negatives are drawn from culturally mismatched examples within the batch.

Let $X_i$ denote an input sequence sampled from culture $c$. We construct its positive pair $X_j$ by selecting another semantically similar sequence from the same culture $c$. For negative pairs, we draw $X_k$ from sequences associated with a different culture $c' \neq c$ within the same batch.

For each representation type $t \in \{\text{explicit}, \text{latent}\}$, we compute the corresponding projection $\mathcal{H}_t^i = f_t(X_i)$, with $f_t$ being the projection function. The contrastive loss is then defined as:

$$\mathcal{L}_{\text{contrast}}^{(t)} = -\log \frac{\exp(\text{sim}(\mathcal{H}_t^i, \mathcal{H}_t^j)/\tau)}{\exp(\text{sim}(\mathcal{H}_t^i, \mathcal{H}_t^j)/\tau) + \sum_{c' \neq c} \sum_{X_k \in c'} \exp(\text{sim}(\mathcal{H}_t^i, \mathcal{H}_t^k)/\tau)}, \tag{2}$$

where $\text{sim}(\cdot)$ denotes cosine similarity and $\tau$ is a temperature parameter. We compute separate losses for each feature type and aggregate them as: $\mathcal{L}_{\text{window}} = \mathcal{L}_{\text{contrast}}^{(\text{explicit})} + \mathcal{L}_{\text{contrast}}^{(\text{latent})}$. This refinement encourages both $\mathcal{H}_{\text{explicit}}$ and $\mathcal{H}_{\text{latent}}$ to organize into structured clusters that reflect intra-group cohesion and inter-group distinctiveness. After training, we obtain the corresponding refined representations say $\widetilde{\mathcal{H}}_{\text{explicit}}$ and $\widetilde{\mathcal{H}}_{\text{latent}}$.

Further theoretical explanations of the contrastive window can be found in Appendix B.2.

## 3.3 Identity Alignment Pool

Cultural reasoning involves identifying symbolic cues and integrating them into a coherent identity that reflects the communicative logic of a cultural context. To this end, we introduce the identity

---

[1] Task semantics reflects the propositional content and domain-specific goals of an utterance. This aligns with Vygotsky's notion of functional meaning as shaped by task context and intentionality [36]. We extract this component from the LLM's standard encoder output, yielding $\mathcal{H}_{\text{task}} = f_{\text{task}}(X)$.

[2] Explicit cultural concepts correspond to overt symbolic forms, such as idioms, honorifics, and role nouns, which encode socially shared meanings and normative structures. Rather than focusing on factual knowledge, named entities, or general commonsense, we extract concepts because they operate at a higher level of abstraction [37]: they organize lexical expressions into culturally salient semantic categories that reflect role expectations, politeness conventions, and institutionalized value systems. This draws on Halliday's systemic functional linguistics [38], emphasizing how culture is made explicit through lexical and phrase-level markers that realize social meaning. We extract this stream using a span infilling objective: $\mathcal{H}_{\text{explicit}} = f_{\text{explicit}}(X)$.

[3] Latent cultural signals capture implicit sentence/discourse level cues (i.e., tone, formality, and indirectness), which modulate the interpretation of meaning across contexts. These are informed by Gumperz's theory of contextualization cues [39], which describe how pragmatic markers convey culturally specific expectations. These signals are modeled via a contextual projection head: $\mathcal{H}_{\text{latent}} = f_{\text{latent}}(X)$.

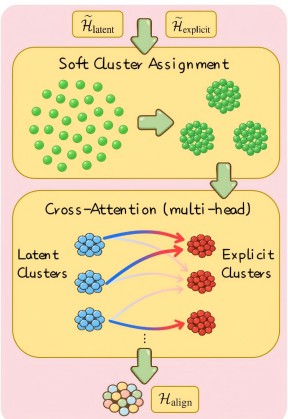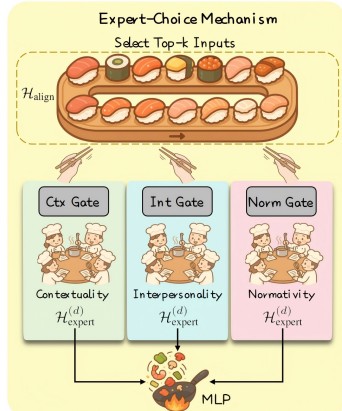

Figure 3: *Left panel*: Cultural feature alignment process. *Right panel*: MoE schematic diagram.

alignment pool, which aligns implicit and explicit cultural features through cross-attention and routes them via an MoE structured around three communicative dimensions (see Figure 3).

To align explicit and latent cultural features, we first summarize them into a set of semantic clusters. Specifically, we apply type-specific Gumbel-Softmax sampling [43] to contrastively refined representations $\widetilde{\mathcal{H}}_{\text{explicit}}$ and $\widetilde{\mathcal{H}}_{\text{latent}}$. Let $\widetilde{\mathcal{H}}$ represent either $\widetilde{\mathcal{H}}_{\text{explicit}}$ or $\widetilde{\mathcal{H}}_{\text{latent}}$ for simplicity. For each row in $\widetilde{\mathcal{H}}$, we compute a soft assignment over $k$ clusters using

$$\mathbf{z} = \text{Softmax}\left((\log \boldsymbol{\pi} + \mathbf{g})/\tau\right), \tag{3}$$

where $\mathbf{g} \sim \text{Gumbel}(0, 1)$, $\boldsymbol{\pi}$ denotes logits produced by a cluster projection head, and $\tau$ is a temperature annealing parameter. We apply this operation separately to both cultural channels:

$$\mathcal{C}_{\text{explicit}} = \text{Cluster}_{\text{gumbel}}(\widetilde{\mathcal{H}}_{\text{explicit}}), \quad \mathcal{C}_{\text{latent}} = \text{Cluster}_{\text{gumbel}}(\widetilde{\mathcal{H}}_{\text{latent}}). \tag{4}$$

Each row in $\mathcal{C}$ represents a soft cluster assignment produced from the corresponding row in $\widetilde{\mathcal{H}}$. This soft clustering technique allows differentiable clustering over heterogeneous cultural signals while preserving fine-grained alignment.

We then apply multi-head cross-attention from latent clusters to explicit clusters, and obtain

$$\mathcal{H}_{\text{align}} = \text{CrossAttn}(\mathcal{C}_{\text{latent}}, \mathcal{C}_{\text{explicit}}), \tag{5}$$

which captures higher-order symbolic-pragmatic consistency across modalities. This mechanism enables culturally linked clusters, such as indirect tone (latent) and honorific markers (explicit), to interact and form integrated pragmatic representations.

To perform cultural specialization, we categorise cultural variation along three high-level communicative dimensions: *Contextuality*, *Interpersonality*, and *Normativity*. These dimensions are not arbitrary; they are theoretically grounded and selected based on (i) their expressivity in linguistic behavior, (ii) their foundational support in cultural communication theory, and (iii) their cross-contextual generalizability in modelling pragmatic variation. The detailed definitions and theoretical motivations of these newly introduced dimensions are provided in Appendices A.1, A.2, and A.3.

While not exhaustive, these three dimensions effectively model language-based cultural variation. Contextuality shapes meaning density and distribution, Interpersonality governs relational stance, and Normativity modulates appropriateness based on internalized values. Together, they span symbolic and pragmatic layers of communication, making them well-suited for integration into an expert-driven reasoning mechanism. Each dimension $d \in \{Contextuality, Interpersonality, Normativity\}$ is associated with a set of experts $\{\mathcal{E}_k^{(d)}\}_{k=1}^{K_d}$, where $K_d$ denotes the number of experts under dimension $d$. Rather than assigning inputs to experts, we adopt an expert-choice mechanism, i.e., each expert actively selects the inputs it specializes in. A detailed theoretical explanation of the expert choice and routing mechanism are provided in Appendix B.3.

Given the aligned cultural representation $\mathcal{H}_{\text{align}}$ obtained in Eqn (5), each expert $\mathcal{E}_k^{(d)}$ first computes a selection score throughout the input sequence, i.e.,

$$s_k^{(d)} = \text{AvgPool}(\mathcal{H}_{\text{align}}) \cdot W_k^{(d)} + b_k^{(d)}, \tag{6}$$

where $W_k^{(d)}$ and $b_k^{(d)}$ are expert-specific gating parameters, and $\text{AvgPool}(\cdot)$ computes the global average of $\mathcal{H}_{\text{align}}$ across the sequence dimension. Each expert then selects the top-$k$ inputs with the highest selection scores and computes the corresponding soft gating weights, i.e., for $k \in \mathcal{I}_d$,

$$\mathcal{I}_d = \text{TopK}(s^{(d)}, k), \;\; \alpha_k^{(d)} = \exp(s_k^{(d)}) / \sum_{j \in \mathcal{I}_d} \exp(s_j^{(d)}), \tag{7}$$

where $\mathcal{I}_d$ is the index set of the top-$k$ experts selected under dimension $d$, and $\alpha_k^{(d)}$ denotes the normalized gating weight for expert $\mathcal{E}_k^{(d)}$. Our inspiration is that not all features can be matched to corresponding cultural dimensions; by allowing each expert to self-select the inputs it best specialises in, we promote soft and competitive specialization among experts.

The gated output for dimension $d$ is computed as a weighted aggregation of the selected expert outputs:

$$\mathcal{H}_{\text{expert}}^{(d)} = \sum_{k \in \mathcal{I}_d} \alpha_k^{(d)} \cdot \mathcal{E}_k^{(d)}(\mathcal{H}_{\text{align}}). \tag{8}$$

Finally, the dimension-specific expert outputs are fused through an MLP and combined with the original refined cultural features via a residual connection to form a unified cultural identity:

$$\mathcal{H}_{\text{self}} = \text{MLP}([\mathcal{H}_{\text{expert}}^{(\text{Ctx})}; \mathcal{H}_{\text{expert}}^{(\text{Int})}; \mathcal{H}_{\text{expert}}^{(\text{Norm})}]) + (\widetilde{\mathcal{H}}_{\text{explicit}} \oplus \widetilde{\mathcal{H}}_{\text{latent}}). \tag{9}$$

Here, $\mathcal{H}_{\text{expert}}^{(\text{Ctx})}$, $\mathcal{H}_{\text{expert}}^{(\text{Int})}$, and $\mathcal{H}_{\text{expert}}^{(\text{Norm})}$ refer to the expert outputs corresponding to the three cultural dimensions of Contextuality, Interpersonality, and Normativity, respectively.

The resulting $\mathcal{H}_{\text{self}}$ encodes a culturally grounded identity state that preserves symbolic structure, pragmatics, and value alignment, enabling CALM to reason in a manner that is sensitive to cultural identity as a structured, multifaceted internal representation.

Further theoretical explanations of the identity alignment pool can be found in Appendix B.4.

## 3.4 Reflective Reasoning Loop

Human cultural intelligence involves retrieving knowledge and reflectively revising behavior [44]. To simulate this metacognitive process, CALM incorporates a reflective reasoning loop, enabling culturally appropriate reasoning and revision of culturally inconsistent outputs. This loop consists of two phases: (i) cultural self-prompted reasoning and (ii) inverse identity calibration, triggered only upon detecting mismatches with the model's internalized cultural identity.

Given the task semantics $\mathcal{H}_{\text{task}}$ and the cultural self-representation $\mathcal{H}_{\text{self}}$ in Eqn (9), CALM first generates a self-prompt $P$ using a lightweight Transformer decoder, i.e.,

$$P = f_{\text{prompt}}([\mathcal{H}_{\text{task}} \oplus \mathcal{H}_{\text{self}}]). \tag{10}$$

This prompt embeds culturally grounded rhetorical framing, stylistic tone, and value-sensitive expressions. For example, in high-context or hierarchical settings, the prompt may emphasize indirectness and deference ("In a hierarchical society, it is customary to..."), whereas in low-context or egalitarian cultures, it may favor clarity and direct self-assertion. The prompt $P$ is prepended to the input sequence $X$ and passed to a decoder $f_{\text{reason}}$ that performs culturally shaped reasoning, yielding

$$\hat{Y}_{\text{raw}} = f_{\text{reason}}([P; X]). \tag{11}$$

This step completes the initial inference. If no cultural inconsistency is detected, the output $\hat{Y}_{\text{raw}}$ is accepted.

To detect cultural misalignment, we adopt a reverse inference strategy that estimates whether the generated output reflects the intended cultural identity. This strategy is likewise grounded in metacognitive models of self-monitoring [45], which emphasize the importance of shifting perspective to evaluate one's own outputs. Specifically, we prompt the base model with

$$\hat{\mathcal{H}}_{\text{reverse}} = f_{\text{identity}}(\hat{Y}_{\text{raw}}), \tag{12}$$

where $f_{\text{identity}}$ is an inverse prompting head that generates a soft distribution or embedding over possible cultural profiles based on the linguistic features of $\hat{Y}_{\text{raw}}$; for classification tasks, it instead

uses intermediate decoder representations prior to prediction. We then compare the reverse-inferred identity $\hat{\mathcal{H}}_{\text{reverse}}$ with the original cultural self-representation $\mathcal{H}_{\text{self}}$ using cosine similarity $\delta = \text{sim}(\mathcal{H}_{\text{self}}, \hat{\mathcal{H}}_{\text{reverse}})$.

If $\delta < \tau$, where $\tau$ is a calibration threshold, we consider the output culturally misaligned. This discrepancy triggers a corrective cycle: CALM reactivates the identity alignment pool to revise $\mathcal{H}_{\text{self}}$, regenerates a new prompt $P'$, and re-enters the reasoning phase.

This self-reflective mechanism enables CALM to perform continual metacognitive correction in dynamic cultural environments, thereby maintaining cultural consistency.

## 4 Experiments

**Tasks:** Following prior survey works [46, 47], we categorize cultural awareness evaluation into two domains: (i) knowledge-oriented, focusing on culturally grounded commonsense reasoning and value reasoning; and (ii) toxicity-sensitive, targeting the detection of culturally harmful content such as hate speech and social bias.

**Datasets:** For commonsense reasoning, we adopt CultureAtlas [48], a fine-grained benchmark spanning over 2,500 ethnolinguistic groups, 193 countries, and 10,000 cities, containing cultural statements labeled as true or false across domains such as festivals, marriage, clothing, food, education, and social behaviors. It further distinguishes between general facts and context-specific assertions (e.g., age, gender, religion), enabling nuanced assessment across different resource levels. For value reasoning, we

Table 1: The memory cost, inference cost, and parameter count of CALM across four datasets.

| Model Params (B) | Inference VRAM (GB) |
|---|---|
| 32.92 | > 67 |

| Dataset | Inference FLOPs/sample (T) |
|---|---|
| UniVaR | 1.29 |
| CultureAtlas | 2.59 |
| CREHate | 0.65 |
| EMGSD | 1.01 |

use the multilingual UniVaR dataset [49], comprising approximately 1 million QA pairs generated by 15 LLMs in 25 languages, covering 87 human values derived from several foundational theories of cultural values [50, 51]. Paraphrased and translated prompts enhance cultural diversity, while answers are back-translated to English to support language-neutral embeddings. For hate speech detection, we employ CREHate [52], a cross-cultural English benchmark consisting of 1,580 social media posts annotated by raters from five English-speaking regions with distinct cultural backgrounds, namely Australia (AU), the United Kingdom (GB), the United States (US), South Africa (ZA), and Singapore (SG). The dataset integrates re-annotated samples from the SBIC corpus [53] and newly curated Reddit and YouTube posts collected using culture-specific hate-related keywords. For social bias detection, we use the EMGSD dataset [54], which contains 57,201 instances labeled for binary and multi-class classification across six demographic dimensions: gender, race, nationality, religion, profession, and LGBTQ+. EMGSD extends the MGSD dataset [55] with subsets from WinoQueer [56] and SeeGULL [57], using GPT-4 and Mistral for additional sentence generation while maintaining human-validated stereotypes annotations.

**Implementation Details:** The implementation details are listed in Appendix C. Table 1 and Figure 4 summarize the training efficiency and computational cost of CALM across datasets.

**Evaluation Metrics:** We evaluate four selected tasks using established frameworks and corresponding metrics: accuracy for hate speech detection, Macro F1 score for social bias detection, accuracy (including Acc@1, Acc@5, and Acc@10) and F1 score for value reasoning, and precision, recall, and F1 score for commonsense reasoning.

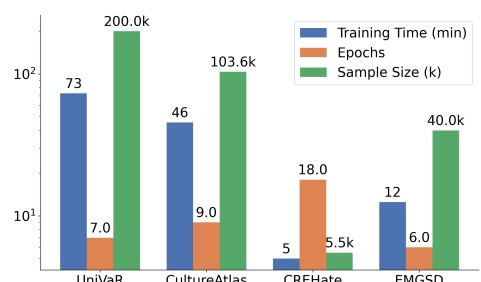

Figure 4: Log-scale statistics of CALM's training time, epochs, and sample size across four datasets.

**Comparative Models:** We follow prior frameworks by adopting the official baselines provided with each benchmark. The selection covers both pretrained encoder models and LLMs, including open-source, instruction-tuned, and proprietary systems. Most baselines are fine-tuned or task-specific,

representing the established state of the art. This consistent setup ensures fairness, continuity, and a rigorous comparison for CALM.

## 4.1 Overall Performance

Table 2: Comparison (%) between CALM and baseline models on the CultureAtlas dataset, evaluated by Precision (P), Recall (R), and F1-score (F).

| Model | All Culture | | | High Resource | | | Mid Resource | | | Low Resource | | |
|---|---|---|---|---|---|---|---|---|---|---|---|---|
| | P | R | F | P | R | F | P | R | F | P | R | F |
| LLaMA-2-7B [58] | 84.2 | 42.1 | 56.1 | 86.8 | 45.6 | 59.8 | 83.3 | 42.9 | 56.6 | 87.0 | 20.7 | 33.5 |
| LLaMA-2-13B | 63.6 | 77.1 | 69.7 | 56.1 | 80.9 | 66.3 | 64.1 | 75.5 | 69.3 | 53.3 | 20.5 | 29.6 |
| Vicuna-7B [59] | 79.6 | 56.8 | 66.3 | 77.3 | 47.2 | 58.6 | 79.4 | 57.9 | 67.0 | 81.3 | 55.7 | 66.1 |
| Vicuna-13B | 67.4 | 81.2 | 73.7 | 68.9 | 81.0 | 74.5 | 69.4 | 82.4 | 75.3 | 67.8 | 82.3 | 74.3 |
| GPT-4 [60] | **95.8** | **90.6** | **93.1** | **95.9** | **91.4** | **93.6** | **94.9** | **92.1** | **93.5** | **94.1** | **90.1** | **92.1** |
| **CALM (Ours)** | 93.6 | 87.7 | 89.1 | 95.0 | 90.9 | 92.4 | 92.5 | 90.3 | 91.2 | 93.2 | 86.3 | 88.6 |

Firstly, Table 2 reports that CALM achieves an F1 score of 89.1% on cultural commonsense reasoning, significantly outperforming all open-source baselines that have undergone human preference alignment. More importantly, CALM demonstrates consistent improvements across high-, mid-, and low-resource cultural groups, while maintaining a strong balance between precision and recall. The definitions of different resource levels are provided in Appendix E.1. This indicates that the model not only adapts well to diverse cultural contexts but also generalizes robustly across cultural strata. Such robustness is particularly critical for low-resource settings, where linguistic coverage is sparse and cultural norms vary more drastically. Despite not relying on additional supervision, CALM achieves performance comparable to the proprietary model GPT-4.

Table 3 assesses the ability of the model to encode and differentiate human values between languages and cultures; CALM achieves the highest scores across all five evaluation metrics, including Acc@1 (23.04%), Acc@5 (50.68%), and Acc@10 (65.41%), outperforming the best UniVaR variant by a clear margin. We report both k-NN and linear probing results to capture complementary aspects of value representation: k-NN evaluates local clustering and proximity of value-aligned responses, while linear probing assesses whether value-relevant dimensions are linearly separable in the embedding space. CALM achieves superior performance on both metrics, indicating that its value encoding is not only semantically coherent but also structurally disentangled from non-value factors. Additional experiments and validations are provided in Appendices H.1 and H.2.

Table 3: Evaluation (%) of CALM and baselines on the UniVaR dataset. k-NN and Linear reflect the quality of cultural embeddings.

| Model | k-NN | | Linear | | |
|---|---|---|---|---|---|
| | Acc | F1 | Acc@1 | Acc@5 | Acc@10 |
| GloVe [61] | 2.27 | 2.26 | 5.45 | 17.19 | 27.72 |
| BERT [62] | 1.78 | 1.82 | 10.57 | 28.87 | 42.20 |
| RoBERTa [63] | 1.88 | 1.89 | 10.06 | 27.70 | 41.17 |
| XLM-R [64] | 1.40 | 1.41 | 8.65 | 24.96 | 37.92 |
| MPNet [65] | 1.40 | 1.49 | 4.73 | 15.74 | 25.80 |
| LaBSE [66] | 4.03 | 3.94 | 11.76 | 32.16 | 47.48 |
| UniVaR | 20.37 | 16.84 | 18.67 | 45.75 | 61.70 |
| **CALM (Ours)** | **23.87** | **21.35** | **23.04** | **50.68** | **65.41** |

As presented in Table 4, CALM attains an average accuracy of 83.66% in cross-cultural hate speech detection, exceeding GPT-4 by more than 5% and substantially outperforming all open-source baselines. Notably, CALM maintains balanced performance across all five countries, including Singapore and South Africa, which have the lowest inter-annotator agreement, demonstrating strong

Table 4: Accuracy (%) comparison across five countries and average (Avg) on CREHate dataset.

| Model | GB | US | AU | ZA | SG | Avg |
|---|---|---|---|---|---|---|
| Orca-2 [67] | 69.99 | 69.09 | 69.80 | 68.80 | 68.61 | 69.26 |
| Flan-T5 [68] | 68.58 | 67.49 | 68.28 | 68.35 | 68.15 | 68.17 |
| OPT [69] | 66.25 | 69.29 | 64.68 | 66.94 | 64.11 | 66.25 |
| GPT-3.5 | 72.47 | 70.62 | 72.39 | 69.28 | 71.94 | 71.34 |
| GPT-4 | 79.66 | 80.64 | 78.02 | 78.03 | 74.65 | 78.20 |
| **CALM (Ours)** | **85.03** | **85.29** | **83.19** | **83.42** | **81.38** | **83.66** |

Table 5: Comparison of CALM and baseline models on the EMGSD dataset.

| Model | Macro F1 (%) | Emissions (g $CO_2$e) |
|---|---|---|
| GPT-4o [70] | 64.8 | Unknown |
| DistilBERT [71] | 80.6 | 156.48 |
| DistilRoBERTa | 53.9 | Unknown |
| ALBERT-V2 [72] | 81.5 | **2.88** |
| BERT | 82.8 | 270.68 |
| **CALM (Ours)** | **85.3** | 158.42 |

generalizability across geographically distinct cultural interpretations within the same language. Additional experiments and validations are provided in Appendices G.1 and G.2.

In the domain of social bias detection, Table 5 shows that CALM reaches a Macro F1 score of 85.3%, surpassing fine-tuned BERT and ALBERT-V2 by 2.5% and 3.8%, respectively. This significant improvement indicates that CALM effectively mitigates group-specific bias while maintaining robust generalization across diverse social contexts. Notably, CALM delivers this performance while maintaining carbon emissions comparable to highly efficient models like DistilBERT, despite achieving significantly higher accuracy, which highlights the sustainability of its design. More impressively, CALM surpasses the proprietary GPT-4o model by over 20%. Additional experiments and validations are provided in Appendices F.1, F.2, F.3, F.4, and F.5.

## 4.2  Ablation Study

Table 6: Comparison of CALM built on various backbone models of different sizes and architectures.

| Backbone | All Culture | | | High Resource | | | Mid Resource | | | Low Resource | | |
|---|---|---|---|---|---|---|---|---|---|---|---|---|
| | P | R | F | P | R | F | P | R | F | P | R | F |
| LLaMA-3.1-8B [73] | 90.5 | 68.9 | 78.1 | 93.4 | 74.2 | 82.6 | 89.2 | 71.7 | 79.3 | 86.1 | 64.4 | 76.1 |
| Gemma-3-12B [74] | 87.0 | 70.3 | 77.6 | 90.7 | 72.5 | 80.4 | 85.6 | 74.2 | 79.4 | 83.9 | 67.2 | 76.2 |
| Gemma-3-27B | 92.2 | 76.4 | 84.6 | 95.1 | 79.1 | 86.7 | 91.0 | 77.8 | 84.7 | 90.3 | 74.6 | 81.8 |
| Qwen3-8B | 89.7 | 67.8 | 76.8 | 92.5 | 73.1 | 81.5 | 88.3 | 69.8 | 77.9 | 86.9 | 62.1 | 72.7 |
| Qwen3-14B | 93.3 | 71.9 | 81.2 | **96.2** | 76.6 | 85.1 | 91.9 | 74.1 | 82.0 | 90.8 | 67.5 | 76.5 |
| **CALM (Qwen3-32B)** | **93.6** | **87.7** | **89.1** | 95.0 | **90.9** | **92.4** | **92.5** | **90.3** | **91.2** | **93.2** | **86.3** | **88.6** |

A potential concern arises from the scaling laws [75], which suggest that model performance often correlates with parameter size. To further examine this, we implemented CALM on a range of backbone models with diverse architectures and parameter scales on the CultureAtlas dataset. As shown in Table 6, the observed improvements originate from CALM's cultural reasoning architecture rather than model scale. This indicates that a well-designed framework, rather than raw model size, is the key to achieving superior performance. The results demonstrate that even when built upon models of comparable or smaller size, CALM consistently and significantly surpasses the official baselines of each benchmark. This finding confirms that CALM exhibits strong generalization and transferability across model families including Qwen, LLaMA and Gemma, at 8B, 14B, 27B, and higher scales. Its robust performance stems from the proposed cultural alignment mechanisms, which provide stability and adaptability independent of the underlying foundation model. In essence, the backbone serves merely as an implementation carrier, while the cultural self-awareness design of CALM is the true driver of its effectiveness.

Table 7 presents further ablation studies on key components, showing that CALM's performance improvements are progressive and interpretable. To assess the contribution of cultural feature streams within the abstract cognitive space, we independently ablate the latent cultural signal and the explicit

Table 7: The ablation study highlights the contribution of each key component to overall performance on the CultureAtlas dataset.

| Component | All Culture | | | High Resource | | | Mid Resource | | | Low Resource | | |
|---|---|---|---|---|---|---|---|---|---|---|---|---|
| | P | R | F | P | R | F | P | R | F | P | R | F |
| Latent Cultural Signal | 91.7 | 85.2 | 87.8 | 93.3 | 88.4 | 90.7 | 90.6 | 87.9 | 89.2 | 90.7 | 82.8 | 86.1 |
| Explicit Cultural Concept | 91.2 | 84.1 | 87.1 | 92.5 | 87.2 | 89.7 | 89.9 | 86.4 | 88.1 | 91.2 | 83.9 | 87.4 |
| Contrastive Window | 91.0 | 83.7 | 86.9 | 92.3 | 87.0 | 89.5 | 89.5 | 85.9 | 87.6 | 90.3 | 82.2 | 85.6 |
| Identity Alignment Pool | 87.3 | 74.2 | 80.1 | 88.0 | 76.4 | 81.7 | 85.6 | 73.6 | 79.1 | 85.2 | 68.9 | 76.3 |
| Replace Cross-Attn with Fusion | 90.4 | 82.1 | 86.2 | 91.5 | 85.5 | 88.3 | 89.1 | 85.2 | 87.1 | 90.2 | 81.3 | 85.1 |
| w/o Cross-Attn | 89.1 | 78.5 | 84.5 | 90.3 | 81.7 | 85.8 | 87.2 | 80.2 | 83.5 | 87.7 | 74.1 | 80.2 |
| w/o MoE | 88.5 | 79.4 | 83.5 | 89.7 | 82.5 | 85.6 | 86.6 | 81.6 | 83.9 | 87.0 | 76.3 | 81.3 |
| w/o Cultural Dimensions | 88.0 | 78.6 | 82.9 | 89.2 | 81.8 | 85.1 | 86.1 | 80.7 | 83.3 | 86.4 | 75.2 | 80.6 |
| w/o Expert-Choice | 88.3 | 78.9 | 83.1 | 89.5 | 82.0 | 85.2 | 86.2 | 81.0 | 83.5 | 86.8 | 75.6 | 80.9 |
| Reflective Reasoning Loop | 89.6 | 81.9 | 85.3 | 91.0 | 84.5 | 87.5 | 88.0 | 83.7 | 85.7 | 88.4 | 79.1 | 83.2 |
| w/o Prompt Generator Head | 90.1 | 81.8 | 85.7 | 91.4 | 84.9 | 88.0 | 88.8 | 84.2 | 86.3 | 89.7 | 80.4 | 84.7 |
| w/o Inverse Identity Calibration | 90.5 | 83.3 | 86.7 | 91.6 | 85.2 | 88.3 | 89.4 | 85.8 | 87.5 | 90.3 | 81.4 | 85.6 |
| **CALM (Ours)** | **93.6** | **87.7** | **89.1** | **95.0** | **90.9** | **92.4** | **92.5** | **90.3** | **91.2** | **93.2** | **86.3** | **88.6** |

cultural concept branches, resulting in overall F1 score reductions of 1.3% and 2.0%, respectively. The removal of the latent cultural signal leads to a particularly pronounced degradation in low-resource settings, with a 2.5% drop in F1, indicating its role in capturing subtle socio-cultural variations that are not explicitly lexicalized. These results confirm that both cultural streams provide complementary signals: the latent stream enhances sensitivity to pragmatics in resource-scarce environments, while the explicit stream anchors interpretation in lexicalized cultural norms.

Removing the contrastive window leads to a 2.2% drop in overall F1, with consistent degradation across high-, mid-, and low-resource settings, indicating that cultural cues, particularly implicit ones, form non-uniform clusters in the embedding space that benefit from contrastive shaping. Without this module, culturally adjacent classes become more confusable. The identity alignment pool yields the largest performance drop among all ablations: a 9.0% overall F1 decrease, and 12.3% in low-resource cultures where explicit cues are limited. These results validate our core hypothesis that effective cultural understanding requires not only extracting symbolic and pragmatic features, but also structurally integrating them to capture the communicative logic of diverse cultural contexts.

We further assess the role of cross-modal alignment by ablating the cross-attention mechanism. Replacement with a shallow MLP that concatenates and projects the latent and explicit streams without token-level interaction yields a 2.9% drop in overall F1, suggesting limited complementary capture but insufficient fine-grained alignment. Fully removing the integration between the streams leads to a 4.6% F1 drop, with pronounced degradation in low-resource settings. These results underscore the importance of token-level alignment in resolving mismatches between stylistic and symbolic cultural cues, particularly when cultural meaning spans multiple linguistic layers.

We evaluate the cultural specialization mechanism within the identity alignment pool by ablating its three core components: expert structure, dimension-specific partitioning, and dynamic routing. Replacing the MoE with a shared MLP results in a 5.6% F1 drop, highlighting the importance of structurally distinct expert pathways for modeling diverse cultural traits. Flattening the dimension-specific organization into a single undifferentiated expert pool further degrades performance, confirming that organizing culture along axes such as Contextuality, Normativity, and Interpersonality improves cross-cultural generalization. Finally, substituting dynamic routing with uniform averaging yields a 6.0% F1 drop, indicating that selective expert activation is critical for cultural compatibility and minimizing representational interference.

Lastly, we assess the impact of reflective reasoning, designed to enable self-correction of cultural interpretations. Removing the entire reasoning loop results in a 3.8% F1 drop, with pronounced effects in mid- and low-resource groups. To isolate component contributions, we ablate the prompt generator and inverse identity calibration heads. Excluding the prompt generator yields a 3.4% drop, underscoring the role of culturally grounded input formulations in initiating effective reasoning. Removing the calibration head leads to a 2.4% decrease, indicating its importance in verifying cultural consistency post-generation, particularly when implicit norms are underspecified.

# 5 Limitation and Discussion

Further discussion of CALM's design and limitation, including ethical considerations, is presented in Appendix D.

# 6 Conclusion

This study proposed CALM, a culturally self-aware language model that integrates culture-based self-representation into its reasoning process. CALM does not rely on external prompts or fixed cultural attributes; instead, it models culture as an internal and adaptive component of reasoning. Through comprehensive evaluations across multiple cultural reasoning tasks, CALM demonstrates strong generalization, cross-cultural robustness, and sustainable performance. Follow-up experiments further validate the model's ability to dynamically adapt to diverse linguistic and cultural contexts. We hope this work will inspire future research on culturally aligned reasoning and encourage the development of language models that engage more responsibly and empathetically with global communities. Future extensions may include modelling cultural dynamics over time, capturing evolving stances in conversational agents, or aligning cultural representations with other modalities such as images, speech, or location metadata.

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

# A  Terminology and Conceptual Definitions

## A.1  Contextuality

*Contextuality* captures the degree to which communicative meaning is encoded implicitly or explicitly. This dimension is grounded in Hall's high-/low-context communication theory [76], a foundational framework in intercultural pragmatics and sociolinguistics. Hall's theory distinguishes cultures by their reliance on shared background knowledge versus explicit verbal information during interaction. High-context cultures tend to prefer indirectness, inference, and omission (e.g., ellipsis or deixis), while low-context cultures rely on self-contained and overt encoding. In our model, this dimension guides expert specialization over discourse-level features such as information redundancy, implicature structures, and referential clarity.

## A.2  Interpersonality

*Interpersonality* reflects how speakers negotiate social alignment, face needs, and interpersonal stance in communication. It builds on Brown and Levinson's politeness theory [77], one of the most widely cited frameworks in linguistic anthropology and pragmatics. The theory formalizes how cultures adopt different politeness strategies, including positive politeness, negative politeness, and off-record indirectness, to manage social distance and power asymmetry. It directly informs our model's treatment of sentence-level features such as directness, mitigation, hedging, and speech act formulation, all of which are critical for culturally appropriate relational framing.

## A.3  Normativity

*Normativity* encodes how internalized cultural values influence what is considered appropriate or offensive in communication. It synthesizes insights from Hofstede's cultural dimensions theory [78] and Schwartz's theory of basic human values [79]. Both are foundational models in cross-cultural psychology and value research, extensively validated across more than 70 national cultures [80]. These theories explain how values such as hierarchy, tradition, and sacredness shape linguistic acceptability, taboo sensitivity, and topic framing. In our model, normativity regulates expert activation based on lexical sensitivity to religious or moral terms, modality preferences, and formality expectations, thereby incorporating value-laden constraints into reasoning.

# B  Theoretical Justifications

## B.1  Abstract Cognitive Space

We examine the sufficiency and validity of the abstract cognitive space (ACS) in modelling cultural complexity from both conceptual and empirical perspectives. Our goal is not to exhaustively capture every dimension of culture, since this is a challenge that no existing model has yet achieved. Instead, we aim to construct a structured and theoretically grounded abstraction that meaningfully organizes and represents the core aspects of cultural cognition.

**Theoretical Motivation:** Culture is inherently complex, involving explicit symbols, implicit norms, values, communicative strategies, and contextual practices. Our approach is inspired by leading studies in sociocultural and cognitive linguistics, which consistently identify three fundamental layers necessary for meaningful cultural modelling: *semantic intent* (propositional content), *explicit cultural concepts* (overt symbolic forms such as idioms, honorifics, or role markers), and *latent cultural signals* (pragmatic, implicit, and discourse-level cues such as tone, indirectness, and stance). This triadic decomposition is grounded in the theories of Vygotsky, Halliday, and Gumperz, reflecting how humans process and infer cultural information at multiple cognitive levels.

**Representation of Cultural Complexity:** Operationally, the ACS in CALM is not a single vector but a structured, multi-channel representation. $\mathcal{H}_{\text{task}}$ encodes domain and task semantics, capturing goal-directed meaning. $\mathcal{H}_{\text{explicit}}$ extracts high-level explicit cultural concepts, focusing on socially salient idioms, honorifics, and role nouns. $\mathcal{H}_{\text{latent}}$ captures implicit, sentence- and discourse-level stylistic features, analyzing cues such as formality, tone, and indirectness. This approach is not simple feature engineering but a learning-based disentanglement developed on large-scale data. It is further refined through joint and contrastive learning to ensure that each stream captures unique and

coherent aspects of cultural variation. At the theoretical level, explicit cultural concepts and latent cultural signals together capture both the breadth and the depth of cultural understanding. Explicit concepts refer to symbols, norms, institutions, and overt pragmatic rules, such as honorifics, role markers, and idiomatic expressions (including role titles like "Doctor" or "Madam," institutional expressions such as Japanese honorifics, and group-identity terms), which reflect cultural dimensions like hierarchy, politeness, and collective identity. Latent signals, in contrast, include pragmatic and communicative features that are less easily defined but vital for cultural cognition. These encompass tone, indirectness, formality, ambiguity, emotional nuance, and politeness strategies, serving as contextualization cues and revealing differences in power distance. For instance, indirect speech is prevalent in high-context cultures, while humor, wordplay, and subtle implications reveal underlying cognitive and communicative styles. Therefore, explicit concepts and latent signals correspond to two complementary cultural dimensions: one encompassing symbols, norms, and institutionalized behavior, and the other involving pragmatics, style, cognition, and interaction. As emphasized in sociolinguistic and cross-cultural pragmatics research, cultural differences manifest not only in explicit symbols but also in deeper communicative styles and cognitive patterns. Both aspects are essential for a comprehensive model of culture.

**Task-Oriented Cultural Features:** Our study focuses on two main categories of tasks. The first category includes knowledge-oriented tasks, which evaluate commonsense and value reasoning within cultural contexts. The second category includes toxicity-sensitive tasks, which assess the detection of culturally harmful content such as hate speech and social bias. The complementarity between explicit cultural concepts and latent cultural signals forms the foundation of our model design for both types of tasks. In knowledge-oriented tasks such as cultural commonsense reasoning and value identification, explicit cultural concepts enable the model to directly capture codified knowledge, including ceremonial language, social roles, and terminology related to festivals or institutions, all of which serve as explicit evidence for commonsense judgments and value assignments. For example, determining whether a festival is unique to a particular culture or whether a title reflects social hierarchy depends on explicit semantic markers. Latent cultural signals, on the other hand, help the model capture stylistic and interactional features that are less formalized but essential for contextual reasoning. Examples include euphemisms, hedging, communicative styles reflecting power distance, and subtle cues indicating group identity. These implicit signals often determine whether a statement is perceived as commonsense or as a shared value within a culture. In toxicity-sensitive tasks such as hate speech and bias detection, explicit cultural concepts include sensitive symbols and stereotypical keywords, such as targeted slurs and discriminatory terms. The model must detect and interpret these high-risk words and their contexts accurately. At the same time, many covert forms of toxicity and social bias are not expressed directly; instead, they are embedded in tone, insinuation, sarcasm, passive aggression, or humor, and are especially common in low-resource or high-context cultures. If a model relies only on explicit features, it will likely miss these hidden forms of toxicity and fail to offer adequate protection for marginalized groups. By combining explicit and implicit channels, our multi-channel modeling approach captures both direct knowledge and symbolic cues, as well as the nuanced cultural risks that arise from style and context. This enables the model to achieve robust performance across both categories of tasks, ensuring comprehensive and culturally sensitive understanding.

**Integration in CALM:** ACS serves as the foundation of cultural reasoning rather than its endpoint. Handling cultural complexity is not the sole responsibility of ACS but the result of the entire CALM framework operating in coordination. Within CALM, ACS provides a theory-driven foundation that separates explicit and implicit cultural information from task semantics, supports modular reasoning, and maintains consistency with established linguistic and AI models. Through contrastive learning and type-specific objectives, the two cultural streams are explicitly distinguished and organized, which enhances both diversity and discriminative power. In the subsequent stages of the framework, ACS serves as the input to the Identity Alignment Pool, where explicit and implicit channels are aligned via cross-attention and further processed by a dimension-specific mixture-of-experts mechanism that encompasses contextuality, interpersonality, and normativity. This integration allows the model to specialize and generalize across communicative dimensions while preserving a coherent representational base. ACS functions not as a static lookup table but as a dynamic cognitive state recalibrated through a reflective reasoning loop. This process enables the model to identify and correct cultural misalignments as they appear in context. Through these integrated mechanisms, CALM is constructed as a holistic system capable of addressing cultural complexity rather than as a set of isolated or independent modules.

## B.2 Contrastive Window

We further elaborate on the theoretical and practical motivations for introducing the contrastive window in CALM. The following discussion integrates complementary perspectives from theoretical grounding, neural representation, and generalization benefits.

**Theoretical Foundation:** Structured and separable cultural representations. Cultural features are not isolated points but tend to form dense and semantically coherent groupings in the embedding space. Linguistic phenomena such as self-deprecating expressions or kinship-related terms often exhibit strong local coherence. This distributional structure is consistent with categorization theory in psychology [81], which posits that human cognition naturally organizes complex concepts into structured groups. Contrastive learning provides a principled mechanism for discovering and reinforcing such group structures. By pulling together representations belonging to the same cultural category and pushing apart those of different categories, it establishes clear cultural boundaries in high-dimensional space. Without this mechanism, models may struggle to capture intra-group homogeneity and inter-group heterogeneity within cultural representations.

**Neural Representation:** Correcting cultural dilution in pretraining. LLMs trained under self-supervised objectives typically optimize for average predictive likelihood across tokens or spans. This objective emphasizes statistical regularities and often dilutes fine-grained cultural distinctions, as models tend to prioritize universal linguistic patterns over culturally specific cues. The contrastive window explicitly clusters both explicit and latent cultural features, enforcing semantic separation within the representation space. This structural encoding allows downstream modules to effectively retrieve by culture and mitigates the loss of cultural variation that may occur during generalization.

**Generalization Benefit:** A structural prerequisite for downstream modules. Modules such as Gumbel-Softmax clustering, cross-attention, and MoE-based expert routing depend on structured and disentangled input representations to operate effectively. Without the contrastive window, cultural features may become entangled and indistinguishable, which weakens the ability of downstream components to make selective decisions. The contrastive window pre-aggregates each cultural cluster, providing spatial organization that enables experts to identify and specialize in relevant cultural dimensions. This process directly enhances routing selectivity and specialization, both of which are essential for accurate cultural reasoning.

## B.3 Expert Choice and Routing Mechanism

The expert choice and routing mechanism in CALM is designed to ensure interpretable and theory-driven specialization across cultural dimensions. Its formulation can be understood from four complementary perspectives: theoretical priors, generation of the routable cultural representation, expert routing and specialization, and the relationship between abstract features and dimensional specialization.

**Theoretical Priors:** Building upon well-established theories in cross-cultural communication, we decompose cultural differences into three complementary communicative dimensions: contextuality, interpersonality, and normativity. Contextuality is grounded in Hall's theory of high- versus low-context communication and relates to information density and omission. Interpersonality draws on Brown and Levinson's politeness theory and captures sentence-level strategies for expressing social stance. Normativity is inspired by Hofstede's and Schwartz's value theories and corresponds to lexical and syntactic preferences associated with social appropriateness and value alignment. These three dimensions jointly define the theoretical space in which linguistic phenomena can be interpreted as learning targets, providing explicit guidance for the downstream expert networks.

**Routable Cultural Representation:** The MoE routing mechanism operates on a unified cultural representation $\mathcal{H}_{\text{align}}$, which is produced through a sequential process. The LLM encoding is first decomposed into task semantics ($\mathcal{H}_{\text{task}}$), explicit cultural concepts ($\mathcal{H}_{\text{explicit}}$), and latent cultural signals ($\mathcal{H}_{\text{latent}}$). The explicit stream is optimized through masked reconstruction to learn idioms, honorifics, and other symbolic concepts that typically appear at the word or phrase level. The latent stream leverages style and value cues such as tone, formality, and indirectness at the sentence or discourse level. To promote cultural separability, contrastive losses are applied to both streams, encouraging representations from the same cluster to move closer while pushing apart those from different clusters. Subsequently, Gumbel-Softmax clustering is used for each stream, and multi-head cross-attention aligns implicit (latent) and explicit clusters. This mechanism enables, for example,

implicit "polite tone" clusters to align with explicit "honorific" clusters. The resulting representation $\mathcal{H}_{\text{align}}$ encodes discourse structure, politeness strategies, and value cues, serving as the input to the expert routing network.

**Expert Routing and Specialization:** For each cultural dimension (contextuality, interpersonality, and normativity), a dedicated expert pool is constructed. Each expert is implemented as a lightweight Transformer layer (hidden size 512, FFN size 2048) initialized according to the Qwen scheme, with weights sampled from $\mathcal{N}(0, 0.02)$ and biases set to zero for stability. The routing follows an "expert choice" principle: within each dimension, every expert scans the entire batch, computes an affinity score for each input, and selects the top-$K$ inputs it is most suited to process. Only these selected inputs are routed through the corresponding expert. The gating scores are softmax-normalized across the chosen inputs, ensuring that activation remains sparse and competitive. To avoid load imbalance, a data-balancing regularization term ensures that all experts are engaged across the batch. The specialized experts thus learn to represent distinct cultural subspaces for their selected inputs. Outputs from all three dimensions are concatenated, passed through an MLP, and then combined residually with the aligned representation to form the unified cultural identity representation $\mathcal{H}_{\text{self}}$.

**Dimensional Differentiation:** Abstract features retain meaningful structure because the model's gradient signals and contrastive constraints continue to enforce separability within high-level representations. In particular, task-driven gradients from downstream objectives, such as value recognition, politeness classification, and taboo detection, provide distinct learning signals for each expert pool, guiding the model to focus on patterns specific to each cultural dimension. Contrastive learning and cross-attention constraints further preserve clear cultural cluster boundaries and ensure coherent alignment between implicit and explicit features within $\mathcal{H}_{\text{align}}$. Sparse activation and load-balancing regularization also support dimensional specialization by ensuring that dispatch loss remains low and that each expert pool maintains even usage, preventing the model from collapsing into a single dominant expert.

## B.4   Identity Alignment Pool

The goal of the Identity Alignment Pool is to provide the language model with a unified and culturally grounded identity representation that reflects the hierarchical nature of human cultural understanding. This design draws upon theories in sociocultural cognition and psycholinguistics, which suggest that deep cultural understanding extends beyond the recognition of isolated cues. It involves the structured integration of explicit symbolic concepts and implicit pragmatic signals into a coherent internal state.

To achieve this, we first apply Gumbel-Softmax clustering to the cultural features extracted through contrastive learning. Both explicit cultural concepts and latent cultural signals are clustered separately using this differentiable mechanism. The core idea is that cultural knowledge should not be modeled as flat, token-level facts but as structured and organized representations. In real-world communication, cultural elements such as honorifics or taboos, as well as pragmatic strategies such as indirectness or power distance, often occur in identifiable yet internally coherent groups. While contrastive learning encourages representations to self-organize into clusters, explicit clustering makes this structure observable and accessible to downstream modules such as cross-modal interaction or expert routing. The differentiability of the Gumbel-Softmax operation allows the entire clustering process to be trained end-to-end with the rest of the model, enabling it to capture both static groupings and dynamic adaptations to each input and task.

Next, we apply multi-head cross-attention between the explicit and latent cultural clusters to generate aligned representations. This mechanism enables the model to connect explicit entities, such as family roles or religious groups, with latent cues such as deferential tone or indirect speech. Cultural understanding fundamentally relies on the integration of symbolic and pragmatic dimensions, as many norms and conventions can only be interpreted correctly when both levels are considered together. Token- or phrase-level features alone cannot capture such higher-order cultural logic. Cluster-level cross-attention allows the model to move beyond isolated fragments and learn abstract co-occurrence patterns, for instance, associations like "religious entity + imperative tone." The use of multiple attention heads facilitates many-to-many mappings between symbolic and pragmatic dimensions, reflecting the richness and variability of cultural phenomena in real-world communication.

To represent the multidimensional nature of cultural variation, the aligned features are organized along three theory-driven cultural dimensions: contextuality, interpersonality, and normativity. For each

dimension, a dedicated expert pool is allocated. Within the routing mechanism, each expert actively scans the entire input batch, computes an affinity score with each input, and then selects only the top-$k$ inputs it is best suited to process. This design assumes that cultural variation follows interpretable axes, with distinct phenomena dominated by specific dimensions. For example, collectivist tendencies in East Asian contexts are often associated with high contextuality, while cross-cultural differences in power distance correspond to the broader dimension of normativity. Assigning each dimension to a separate expert pool reduces interference and encourages specialization. This design is inspired by the psychological principle of the division of cognitive labor [82], which posits that specialized units in both cognitive and social systems lead to more efficient and robust reasoning. Routing by expert choice within each cultural dimension outperforms conventional token-level routing by allowing experts to focus on culturally relevant patterns, reducing competition, and promoting the emergence of meaningful expert clusters. Moreover, enabling each expert to select only its top-$k$ inputs rather than processing all inputs reinforces specialization and minimizes redundancy.

Finally, we employ an MLP to integrate the outputs from all cultural dimensions, with residual connections applied to preserve salient cultural signals from the original representations. This structure allows the model to incorporate high-level reasoning from dimension-specific experts while retaining foundational cultural features. The residual mechanism, inspired by architectures such as ResNet and Transformer, prevents the loss of valuable information during deep fusion. Compared with simple averaging or weighted summation, concatenation followed by an MLP supports nonlinear integration of heterogeneous expert outputs, enabling more complex cultural interactions. The resulting identity representation captures both the breadth and depth of cultural knowledge, providing a strong foundation for reflective and culturally sensitive reasoning throughout the CALM framework.

## C  Implementation Details

We use Qwen3-32B [83] as the backbone. Unless otherwise specified, all MLP-based projection heads are implemented as 2-layer networks with hidden size 512, ReLU activation, and a dropout rate of 0.1. The explicit concept stream is trained with a span-infilling objective, where idioms, honorifics, and culturally salient role expressions are randomly masked by sentinel tokens and reconstructed autoregressively. The latent signal stream is trained on a sentence-level cultural style and value classification objective, conditioned on stylistic cues such as tone, formality, indirectness, and inferred cultural stance. Weak cultural group labels derived from language, region, or community metadata are used to supervise this component.

The contrastive window applies contrastive learning separately to the explicit and latent channels using independently parameterized MLP projection heads. Positive pairs are constructed by sampling semantically similar sentences from the same cultural label (e.g., country or language group), while negative pairs are drawn from culturally mismatched examples within the same batch. We use the NT-Xent loss with temperature $\tau = 0.07$ and batch size 64.

In the identity alignment pool, Gumbel-softmax clustering ($K = 5$) is applied to both cultural streams, with temperature cosine-decayed from 1.0 to 0.2. The cluster projection head's hidden size is 256. Multi-head cross-attention ($h = 8$) is computed from latent to explicit clusters. Each communicative dimension contains four experts, each implemented as a 2-layer transformer block with hidden size 512 and FFN size 2048. To avoid expert collapse and promote balanced usage, we apply sparse dispatch loss with top-$k$ activation ($k = 2$) and a load balancing regularization term [84].

The prompt generator is implemented as a lightweight 6-layer causal decoder with hidden size 768, rotary positional encoding, and gated residual connections. This module is trained independently using teacher forcing to autoregressively generate culturally grounded natural language prompts from the joint semantic and identity representation. The inverse identity head is implemented as a 2-layer MLP classifier that produces a softmax distribution over cultural group labels. It is trained using cross-entropy loss on retained cultural supervision signals such as language or regional tags.

The final prediction layer is a task-specific classifier, trained using cross-entropy loss depending on the downstream objective. For classification tasks such as hate speech detection and stereotype recognition, CALM directly predicts the task label from the semantically grounded representation.

We allow at most one reflective cycle. We use a temperature of 0.7 for reasoning generation to encourage culturally diverse outputs. For identity calibration, deterministic decoding with temperature

0.0 is used to ensure consistency and reproducibility. All trainable modules are optimized using AdamW with a learning rate of $3 \times 10^{-5}$, weight decay of 0.01, and linear warmup over the first 10% of training steps. Reported results are averaged over ten runs. All experiments were conducted on an NVIDIA H200 GPU cluster.

# D   Limitation and Discussion

**Use of Technical Terms:** Throughout this paper, we use terms such as "construct," "understand," and "reflect" to describe model-internal operations over cultural representations. These expressions are not meant to imply the model possesses human-like cognitive abilities, but rather to emphasize CALM's distinction from existing approaches. Unlike prior methods that treat culture as an external label or a prompt-level modifier, CALM operationalizes cultural understanding internally, using structured reasoning to integrate, monitor, and adjust cultural representations during inference. This internalization reflects the model's design goal to handle culture as a dynamic reasoning dimension rather than a fixed external attribute.

**Cultural Complexity:** We recognize that culture is not a static entity, but a dynamic and multifaceted system shaped by history, interaction, and context. We also acknowledge that no existing approach can fully capture the vivid, evolving, and layered nature of human cultural identity. While CALM abstracts certain prominent communicative dimensions of culture to enable stable operational modelling, its goal is not to fully replicate the fluidity of human cultural experience, but to support adaptive reasoning over structured cultural representations. By embedding cultural modelling into the internal reasoning process of language models, CALM takes a principled step toward the development of more culturally aware AI systems.

**Evaluation Framework:** A major gap in current culturally aware tasks lies in the absence of a universal evaluation framework. Existing benchmarks often overrepresent individual cultural traits or focus narrowly on specific groups, while a unified standard for evaluating diverse cultural dimensions across populations remains lacking. As previously discussed, the complexity of culture poses significant challenges to the development of a truly comprehensive multicultural evaluation paradigm. We acknowledge that our current work is also constrained by this limitation and look forward to future efforts that may address it.

**Limitations and Ethical Considerations:** While CALM introduces structured cultural modelling, it inherits common LLM-based limitations. First, it relies on cultural signals extracted from large corpora, potentially encoding biases or reinforcing stereotypes. Although our identity alignment balances implicit and explicit signals, it cannot fully prevent amplifying dominant norms or marginalizing underrepresented groups. Second, predefined cultural dimensions, despite dynamic selection, may oversimplify cultural diversity, risking essentialist assumptions [85], such as generalizing "collectivist" or "hierarchical" behaviors. Finally, though CALM simulates reflective reasoning, it lacks genuine cultural understanding, lived experience, or ethical deliberation, and should not be interpreted as a cultural classifier, advisor, or decision-maker in sensitive contexts.

# E   Additional Experiments on the CultureAtlas Benchmark

## E.1   Resource-level Definition

The CultureAtlas benchmark dataset classifies cultural groups into high-resource, medium-resource, or low-resource categories based on two main criteria: (1) the availability of linguistic and digital resources, and (2) the overall level of socioeconomic development associated with each group.

Specifically, high-resource groups (e.g., the United States, China, France, Spain, Japan) are characterised by abundant training data, broad population coverage, and advanced digital infrastructure. Medium-resource groups (e.g., Türkiye, Egypt, Iran, Malaysia, Argentina) exhibit moderate levels of data availability and representativeness. Low-resource groups (e.g., Laos, Bhutan, the Democratic Republic of the Congo, Serbia) refer to cultural communities with very limited data coverage. These groups are often underrepresented or marginalised, either due to economic constraints or because they use minority or poorly documented languages.

# F  Additional Experiments on the EMGSD Benchmark

## F.1  Cross-model Cultural Bias Evaluation

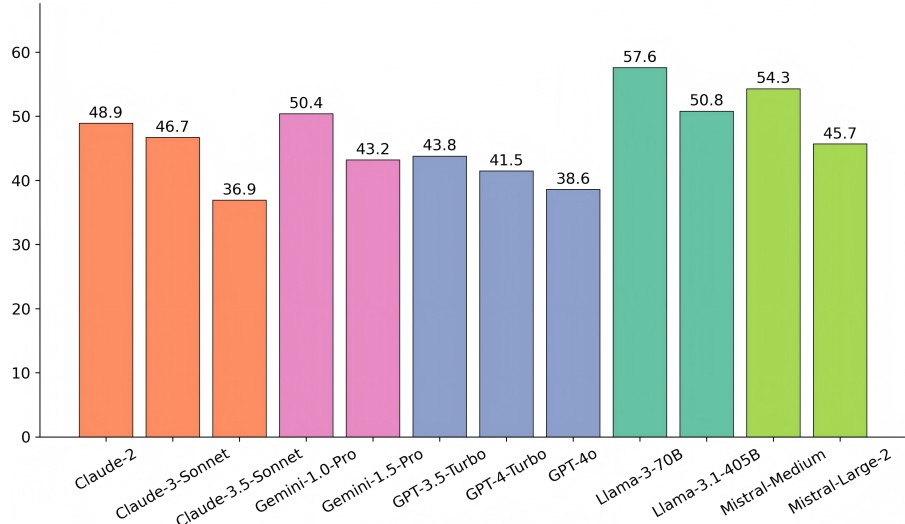

Figure 5: Proportion of stereotypical responses generated by each model on 1,000 culturally neutral prompts derived from the EMGSD benchmark.

To examine the presence of unintended cultural bias in LLMs, we evaluate twelve widely used LLMs by quantifying the proportion of stereotypical responses they generate when prompted with 1,000 culturally neutral sentences derived from the EMGSD benchmark. Formally, for a model $M$, the stereotype prevalence is defined as:

$$P_M = \frac{1}{n} \sum_{i=1}^{n} \mathbf{1}(\hat{y}_i = 1), \tag{13}$$

where $n$ denotes the total number of generated responses and $\hat{y}_i \in \{0, 1\}$ represents the binary prediction of stereotype presence produced by our CALM framework.

As illustrated in Figure 5, the results reveal substantial variation across model families. Claude-3.5-Sonnet exhibits the lowest proportion of stereotypical outputs (36.9%), followed by GPT-4o (38.6%) and Gemini-1.5-Pro (43.2%), indicating stronger alignment with culturally neutral intent. In contrast, models from the LLaMA and Mistral families produce markedly higher proportions of stereotypical responses, with LLaMA-3-70B and LLaMA-3.1-405B reaching 57.6% and 50.8%, respectively.

These findings demonstrate that even state-of-the-art LLMs still reproduce stereotypical or culturally sensitive content under neutral prompting conditions. Such disparities highlight persistent cross-family differences in cultural safety and underscore the importance of enhancing cultural awareness within LLMs to mitigate unintended bias and ensure culturally appropriate behaviour.

## F.2  Demographic-level Analysis of Stereotypes

To further examine how stereotypical content is distributed across different social dimensions, we categorize model responses by demographic group as defined in the EMGSD benchmark and compute the average stereotype proportion per category, as detected by CALM. Figure 6 presents the mean stereotype proportions across six major groups: gender, profession, nationality, race, religion, and LGBTQ+. We observe the highest prevalence of stereotypical content in responses related to profession (79.7%), followed by nationality (52.5%), religion (50.4%), and race (48.3%). In contrast, the lowest bias levels are found in the gender (34.2%) and LGBTQ+ (14.1%) categories.

These findings indicate that stereotype risks are not uniformly distributed but instead vary substantially across demographic backgrounds. Importantly, stereotypical patterns are observed across all

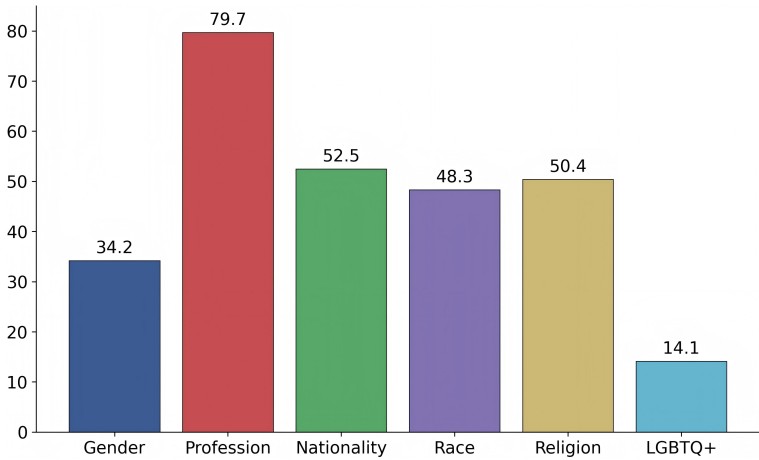

Figure 6: Mean proportion of stereotypical responses by social group, aggregated across all evaluated models. The proportions are computed using the CALM framework on 1,000 culturally neutral prompts derived from the EMGSD benchmark.

demographic groups, suggesting that cultural bias is both pervasive and multifaceted. Even small proportions of stereotypical responses can have disproportionate social impact when models are deployed in sensitive contexts. Such disparities underscore the necessity of fine-grained, group-specific evaluation to ensure the cultural robustness and ethical reliability of LLMs, and further highlight the importance of enhancing cultural awareness to mitigate these biases and foster socially responsible model behaviour.

## F.3 Group-wise Validation of CALM

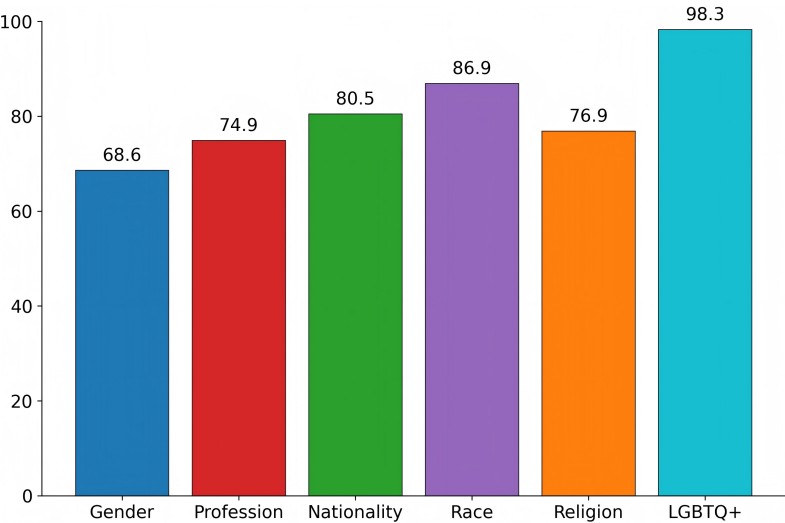

Figure 7: Macro F1 scores of CALM across six demographic groups on the EMGSD test set.

To further validate CALM's bias detection capability under demographic variation, we conduct a fine-grained evaluation of its classification performance across six social groups on the EMGSD test set. Figure 7 reports the macro F1 scores obtained for the "Gender", "Profession", "Nationality", "Race", "Religion", and "LGBTQ+" categories. The results show that CALM maintains consistently strong detection accuracy across all groups, with the highest scores observed for "LGBTQ+" (98.3%) and "Race" (86.9%) categories, where stereotypical expressions are typically more explicit and lexically identifiable. Relatively lower but still robust performance is achieved for "Gender" (68.6%) and "Profession" (74.9%), which often involve subtler, context-dependent linguistic cues. These findings

confirm the reliability of CALM's performance and its ability to generalise effectively across diverse demographic contexts, reinforcing its robustness and interpretability for fine-grained stereotype detection. Such consistency further substantiates the preceding cross-model and group-level analyses, demonstrating that CALM provides a stable and culturally aware foundation for evaluating bias in LLMs.

### F.4 Method-level Analysis of Attribution Consistency

Table 8: Similarity between SHAP and LIME attribution vectors for CALM across 1,000 EMGSD test instances.

| Metric | Mean (Std. Dev.) | $p$-value | Interpretation |
|---|---|---|---|
| Cosine Similarity | 0.672 (0.251) | $< 0.001$ | Moderate alignment |
| Pearson Correlation | 0.639 (0.266) | $< 0.001$ | Moderate alignment |
| Jensen-Shannon Divergence | 0.224 (0.101) | $< 0.001$ | Low divergence |

To evaluate the internal consistency of CALM's token-level explanations, we compare the attribution vectors produced by SHAP and LIME across 1,000 text instances sampled from the EMGSD test set. SHAP (SHapley Additive exPlanations) attributes each token's contribution by estimating its marginal effect on the model's output through all possible feature combinations, while LIME (Local Interpretable Model-agnostic Explanations) perturbs the input text and fits a local surrogate model to approximate token-level importance. For each instance, we generate a pair of attribution vectors $(\phi_i, \beta_i)$ using SHAP and LIME respectively, and compute three similarity metrics: cosine similarity, Pearson correlation, and Jensen-Shannon (JS) divergence. These metrics quantify the alignment of token importance distributions across explanation methods.

We compute the mean similarity $\overline{M}$ and sample standard deviation $s_M$ across the $K = 1000$ samples as follows:

$$\overline{M} = \frac{1}{K} \sum_{i=1}^{K} M(\phi_i, \beta_i), \tag{14}$$

$$s_M = \sqrt{\frac{1}{K-1} \sum_{i=1}^{K} \left( M(\phi_i, \beta_i) - \overline{M} \right)^2}. \tag{15}$$

To determine whether the observed mean similarity is statistically significant, we perform a one-sample $Z$-test against the null hypothesis of no similarity. The $Z$ statistic and two-tailed $p$-value are computed as:

$$z = \frac{\overline{M} - T_M}{s_M / \sqrt{K}}, \tag{16}$$

$$p = 2 \times P(Z > |z|). \tag{17}$$

Here, $T_M$ denotes the theoretical threshold of no similarity: $T_M = 0$ for cosine similarity and Pearson correlation, and $T_M = 1$ for JS divergence. A low $p$-value indicates that the observed similarity is significantly different from the null hypothesis baseline.

As shown in Table 8, CALM demonstrates statistically significant consistency between SHAP and LIME across all metrics. The average cosine similarity (0.672) and Pearson correlation (0.639), together with a low Jensen-Shannon divergence (0.224), indicate that CALM produces coherent and method-agnostic token-level explanations. The observed standard deviations ($\sim$0.25) reflect natural variance across instances, suggesting that the explanation alignment is stable but not artificially uniform. These results support the conclusion that CALM maintains robust internal attribution consistency across perturbation-based and model-based explanation paradigms.

### F.5 Token-level Analysis of Attribution Consistency

We further examine the visual and semantic alignment of token-level attributions between SHAP and LIME. Figure 8 presents a joint visualization of the top 50 tokens ranked by their average attribution

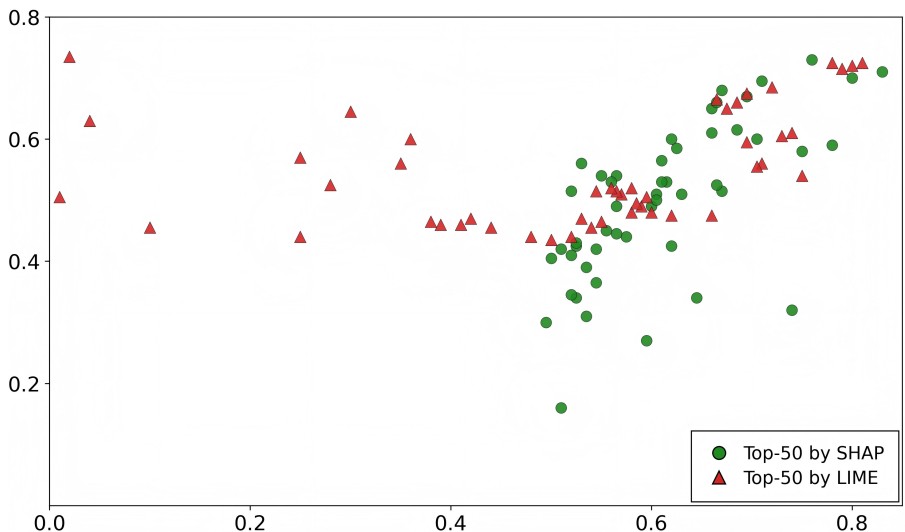

Figure 8: Combined visualization of the top 50 tokens ranked by average SHAP and LIME attribution values using CALM over the EMGSD test set.

scores across both methods over the EMGSD test set. Each token's position corresponds to its mean SHAP score on the $x$-axis and its mean LIME score on the $y$-axis, providing an intuitive view of attribution consistency between perturbation-based and model-based interpretability frameworks. Tokens located near the diagonal line exhibit strong cross-method agreement, whereas those deviating from it indicate attribution divergence between SHAP and LIME. Consistent with the quantitative findings, the visualization reveals a high degree of overlap in tokens that both methods identify as strongly influential.

Representative examples such as "terrorist", "criminals", and "sexist" consistently receive high positive attributions under both methods, reflecting culturally loaded or discriminatory semantics that bias model predictions toward stereotypical interpretations. Other examples such as "femme", "nerdy", and "fanatical" show moderate but aligned attributions, illustrating subtler social stereotypes embedded in gendered or intellectual discourse. Overall, these results demonstrate that CALM captures semantically coherent and cross-method consistent attribution patterns, reinforcing the stability and interpretability of its cultural reasoning process.

## G  Additional Experiments on the CREHate Benchmark

### G.1  Cultural Robustness on Consensus Samples

Table 9: Comparison of CALM with baselines on posts with unanimous in-country agreement from the CREHate dataset.

| Model | GB | US | AU | ZA | SG | Avg |
|---|---|---|---|---|---|---|
| GPT-4 | 94.29 | 95.25 | 93.54 | 92.82 | 87.11 | 92.60 |
| GPT-3.5 | 85.22 | 82.60 | 85.41 | 83.68 | 85.09 | 84.40 |
| Orca-2 | 82.56 | 81.89 | 82.35 | 82.76 | 80.02 | 81.92 |
| Flan-T5 | 82.22 | 80.58 | 80.91 | 81.03 | 81.20 | 81.19 |
| OPT | 77.76 | 80.99 | 76.44 | 77.62 | 74.95 | 77.95 |
| **CALM (Ours)** | **94.57** | **95.31** | **94.13** | **93.49** | **92.76** | **94.05** |

We aim to examine cultural robustness under a high-confidence setting that minimises annotation noise. In the CREHate benchmark, different countries may disagree on whether a post is hateful. To obtain reliable cultural signals, we focus on the subset in which all annotators within each country reached full agreement on the label. Each column in Table 9 therefore represents a country-specific normative judgement, allowing us to evaluate whether a model can align with culturally grounded majority expectations rather than relying on ambiguous or disputed examples. For each country (GB,

US, AU, ZA, SG), we retain only the test samples that received unanimous in-country agreement and use these as gold-standard references. All models are evaluated under identical conditions, and accuracy is reported as the main metric. This controlled design ensures that any remaining performance difference reflects the model's intrinsic cultural robustness rather than inconsistencies in human annotation.

Our CALM achieves the highest accuracy across all five countries, with an average of 94.05%. The performance remains consistently strong across culturally diverse regions, including Singapore and South Africa, where human consensus is usually harder to reach. CALM not only exceeds the best-performing proprietary baseline in overall accuracy but also exhibits a much smaller variance across countries, indicating its balanced adaptation to distinct cultural contexts. These findings demonstrate that CALM effectively captures culturally grounded cues and maintains stable, identity-sensitive predictions across diverse English-speaking contexts. Its consistent accuracy in both culturally homogeneous and heterogeneous societies, such as Singapore and South Africa, indicates that the model has internalised broader cultural reasoning rather than overfitting to region-specific linguistic patterns. Overall, CALM achieves superior accuracy, fairness, and cross-cultural stability, confirming its strong capacity for culturally informed language understanding.

### G.2 Evaluation of Instruction Adherence

Table 10: Comparison of CALM with baseline models in terms of out-of-choice (OOC) rates on the CREHate binary prompt task.

| Model | OOC Rate (%) |
|---|---|
| GPT-4 | 0.09 |
| GPT-3.5 | 0.01 |
| Orca-2-7B | 0.00 |
| Flan-T5-XXL | 0.00 |
| OPT | 0.11 |
| **CALM (Ours)** | **0.00** |

Generative models occasionally fail to produce answers in the required format (e.g., "a", "b", "hate", or "non-hate"), a phenomenon referred to as out-of-choice (OOC). OOC responses indicate a model's inability to strictly follow the prompt instruction, such as when it outputs a full-sentence explanation or an unrelated completion instead of a predefined choice. Under the CREHate binary prompt setup, we evaluate the proportion of such outputs to measure instruction-following reliability, which can be formally defined as:

$$\text{OOC} = \frac{1}{N} \sum_{i=1}^{N} \mathbb{I}[\hat{y}_i \notin \mathcal{Y}_{\text{valid}}], \tag{18}$$

where $\hat{y}_i$ denotes the model output for input $x_i$, $\mathcal{Y}_{\text{valid}}$ is the predefined set of permissible answer choices (e.g., hate or non-hate), and $\mathbb{I}[\cdot]$ is the indicator function. The metric thus quantifies the proportion of responses that violate the instruction format.

Table 10 reports the OOC rates for all compared models. Instruction-tuned generative models like GPT-4 and GPT-3.5 achieve relatively low OOC rates, yet they still occasionally violate the format constraint by producing elaborated or hedged responses. In contrast, our model CALM records an OOC rate of zero, yielding perfectly valid labels for all inputs. While CALM retains full generative capacity, its prompt alignment and deterministic decoding strategy naturally constrain generation to task-relevant tokens without any explicit vocabulary masking. This design ensures complete adherence to prompt specifications, independent of linguistic variation or cultural context, and demonstrates CALM's superior reliability and stability in instruction-based prediction tasks.

## H Additional Experiments on the UniVaR Benchmark

### H.1 Cross-cultural Value Generalization

Ensuring a balanced representation of diverse value systems is critical to prevent overfitting to particular cultural dimensions or normative structures. Our training process does not rely on direct

Table 11: Value identification accuracy (%) across four benchmarks with k-NN and Linear probing.

| Model Name | WVS | | PVQ-RR | | GLOBE | | ValuePrism | |
|---|---|---|---|---|---|---|---|---|
| | k-NN | Linear | k-NN | Linear | k-NN | Linear | k-NN | Linear |
| GloVe | 1.31 | 4.25 | 3.11 | 5.82 | 2.49 | 3.72 | 2.18 | 8.00 |
| BERT | 1.15 | 8.57 | 2.99 | 11.34 | 1.88 | 7.45 | 1.11 | 14.92 |
| RoBERTa | 1.36 | 7.82 | 2.83 | 10.94 | 1.95 | 6.99 | 1.39 | 14.51 |
| XLM-R | 0.75 | 7.12 | 2.53 | 8.85 | 1.56 | 6.23 | 0.76 | 12.38 |
| MPNet | 0.83 | 4.36 | 1.75 | 4.83 | 1.49 | 2.86 | 1.51 | 8.47 |
| LaBSE | 2.44 | 9.97 | 5.99 | 11.55 | 3.61 | 9.31 | 4.08 | 16.20 |
| UniVaR | 21.10 | 19.14 | 17.53 | 16.34 | 21.34 | 18.66 | 21.51 | 20.55 |
| **CALM (Ours)** | **23.31** | **21.62** | **19.90** | **18.66** | **23.36** | **20.91** | **23.71** | **22.57** |

supervision from any single value taxonomy. Instead, it elicits value-relevant behaviour through multi-view contrastive learning on culturally grounded question–answer pairs, followed by translation into a shared language space to minimise linguistic bias. The model learns a compact embedding that focuses on value-salient rather than stylistic information. The four evaluation corpora, WVS [86], PVQ-RR [87], GLOBE [88], and ValuePrism [89], differ substantially in their conceptual scope and structural assumptions. WVS and PVQ-RR follow formalised survey taxonomies. GLOBE focuses on societal and organisational cultural practices. ValuePrism emphasises pluralistic human values, rights, and duties. As shown in Table 11, CALM achieves consistently strong performance across all four benchmarks, showing stable behaviour despite their distinct modelling paradigms. These results remain consistent across languages and cultural settings, indicating that the model's behaviour is largely unaffected by linguistic variation. The stability observed across heterogeneous corpora suggests that CALM effectively distinguishes value-related semantics from other contextual or lexical factors that are unrelated to the underlying cultural meaning. In contrast with systems that depend on direct value labels or handcrafted taxonomies, CALM demonstrates structural abstraction and cultural transfer through semantically grounded representation learning.

## H.2 Cross-domain Linguistic Robustness

Table 12: Cross-domain generalization accuracy (%) on text-only and paraphrase tasks.

| Model Name | text-only | | paraphrase | |
|---|---|---|---|---|
| | Acc@1 | Acc@5 | Acc@1 | Acc@5 |
| GloVe | 12.34 | 63.44 | 13.75 | 65.59 |
| BERT | 17.22 | 66.84 | 26.97 | 72.63 |
| RoBERTa | 15.20 | 66.76 | 19.98 | 69.93 |
| XLM-R | 17.59 | 67.37 | 19.60 | 70.40 |
| MPNet | 15.33 | 65.85 | 26.73 | 72.13 |
| LaBSE | 14.66 | 63.08 | 25.95 | 72.44 |
| UniVaR | 8.33 | 58.73 | 16.73 | 63.16 |
| **CALM (Ours)** | **7.95** | **56.80** | **15.98** | **62.17** |

To further examine whether the learned representation relies on translation artefacts or stylistic regularities, we evaluate cross-domain generalisation using two settings derived from multilingual translation corpora. The first setting, text-only, provides English sentences without any contextual prompts, while the second, paraphrase, wraps each sentence within a question–answer style template to match the structure used during training. As shown in Table 12, traditional sentence-embedding models display higher sensitivity to the source-language signal and thus achieve stronger performance in both settings. By contrast, CALM yields lower scores on this diagnostic task, which confirms that its embedding space is less affected by translationese patterns or surface linguistic traces. This behaviour is consistent with the broader multilingual evaluation, showing that CALM exhibits minimal dependence on specific languages and preserves coherent representations across cultural groups. The consistent outcomes across both settings indicate that CALM maintains stable and conceptually grounded embeddings even when linguistic traces or stylistic variations are intentionally introduced. Together, these results confirm that CALM generalises value semantics rather than encoding superficial language-dependent correlations.

