# OpenReview forum: "CALM: Culturally Self-Aware Language Models"
_NeurIPS.cc/2025/Conference — NeurIPS 2025 poster_

### Official Review · Reviewer_QL7m · 2025-06-29

**Clarity:** 3
**Significance:** 2
**Originality:** 2
**Rating:** 3
**Confidence:** 4

**Summary:**

This paper highlights the LLMs’ potential cultural conflict and sensitivity under different cultural scenarios. It poses an approach to enable LLMs’ deeper understanding of the inputs’ cultural context by embedding cultural features to prompt generation with calibrated identity, thereby enhancing the appropriateness of the output to conform to that cultural context. The performance of the proposed framework is then validated based on different tasks and ablation tests.

**Questions:**

Q1: I am confused with the concept of expert choice and routing mechanism. How can the experts specialize in these dimensions (contextuality, interpersonally, and normativity)? Could you explain more on how the expert is initialized for the three dimensions at their meaning (Line 153-155) and how the Ctx/Int/Norm gate works? How do the dimensions match with the highly abstract features?

Q2: The baseline result in Table 1-4 is taken from the papers [52]-[56] respectively to show that CALM achieves almost the best performance across various types of datasets. As stated in the Appendix, the backbone model of CALM is Qwen-32B. Since CALM is implemented based on a 32B large model, do the favorable results stem from the underlying model itself? For example, in Table 1, the scale of baseline models are 7B and 13B, which are much smaller than CALM and are expected to have poorer results. It would be more compelling if the author could provide some experiment results based on different underlying models that have a relatively same scale as other baseline models.

S1: The use of footnotes is nonstandard. The content length of footnotes in P3 and P5 is already comparable to or even exceeds that of the main text. It is recommended to incorporate key information into the main body and place non-essential explanations in appendices.

S2: Irregular use of bold text in tables. In terms of font style in Tables 1-4, bold carries different meanings. If bold is used to indicate the best-performing entry, then should the "Emissions" column in Table 4 be bolded for ALBERT-V2?

**Ethical Concerns:**

["NO or VERY MINOR ethics concerns only"]

**Final Justification:**

I appreciate the authors' response and clarifications. While their responses were helpful, I find it difficult to fully agree with the statement in their rebuttal: 'The ACS provides a minimal, theory-driven foundation that separates explicit and implicit cultural information...' and their assertion that the entire framework functions effectively. My primary reservation stems from the data-dependent nature of the CALM framework. This inherent dependency presents significant challenges when handling the inherent complexity of cultural phenomena. Consequently, I retain some reservations about the overall coverage and robustness claimed for the framework.

**Limitations:**

Yes

**Quality:**

2

**Strengths And Weaknesses:**

Strengths:
1. This paper addresses an essential issue in LLMs’ awareness of cultural context.
2. The proposed methods are novel and sound. They skillfully incorporated cultural context into cue word generation.
3. The ablation test is clear.

Weakness:
1. Important information is missed in the main body, such as the experiment setting in the Appendix and the very long footnotes. Also, as the paper suggests a complicated network, a more detailed figure or table of the network structure to address these components’ details will be helpful.
2. The experiment of CALM under different backbone models is missing but seems important for the method's effectiveness. It will be better to conduct a comparison between the underlying LLM with and without CALM.
3. The related work section can be improved by adding relevant articles on the key techniques adopted to make the novelty in terms of techniques clearer, such as contrastive learning, MoE, etc.

---

> ### Author Rebuttal · Authors · 2025-07-30
>
> First of all, we sincerely appreciate your valuable feedback and thoughtful comments.
>
> **Q1:** Important information is missed in the main body, such as the experiment setting...
>
> **A1:** Due to page limitations in the main text, we included the experimental settings in Appendix C. This is a standard academic writing practice, frequently adopted in many top-tier venues, including NeurIPS [1, 2]. We chose to place the experimental details in the appendix because we consider the main contributions, including the design, methodology, and theoretical innovations, to be of greater importance and deserving of priority in the main body. This decision is also consistent with NeurIPS’s policy of allowing authors to provide appendix. Moreover, as explicitly indicated in line 225 (“The implementation details are listed in Appendix.”), readers interested in experimental setup and reproducibility can easily locate the necessary information. Therefore, regarding the reviewer’s concern, we would like to clarify that our approach fully complies with academic conventions and NeurIPS guidelines. It is simply a matter of writing style and space optimization. Nonetheless, we value the reviewer’s suggestion and are willing to consider moving implementation details into the main text on additional page in a future revision.
>
> **Q2:** ...a complicated network, a more detailed figure or table...will be helpful
>
> **A2:** We fully agree that clear figures and tables are important for understanding our work. In fact, we have taken steps in this direction. For example, on the right side of Figure 2, we provide a detailed illustration of the expert selection mechanism, as the cultural specialization across communicative dimensions is a major innovation in our paper. To make the workflow clearer and reduce the reading burden, we have visualized this process. Additionally, our ablation table (Table 5) presents comprehensive quantitative results for this complicated network, complemented by thorough qualitative analysis. The architecture diagram in Figure 2 also presents the overall design and detailed workflow of our framework. Note that, some engineering modules, such as Gumbel-Softmax and SimCLR, are well-established components in the literature; providing separate detailed figures for these is not within the scope of our work, but we have cited the relevant original papers appropriately (e.g., lines 118 and 136). Although these components are part of our design, their technical details are already thoroughly documented in the original works, and interested readers can refer to those sources for further information. If you have any specific requests regarding figures or tables that would enhance clarity, we welcome your suggestions and will take on board.
>
> **Q3:** The related work section can be improved by adding...such as contrastive learning, MoE, etc.
>
> **A3:** Contrastive learning and MoE are indeed key techniques in our design. However, the central theme of our paper is “cultural awareness.” In the related work section, we organized the literature according to the two mainstream approaches currently used in this area, as our primary aim is to provide readers with a clear understanding of progress on cultural awareness itself. This is a relatively novel research topic, and as discussed in lines 77–88, most existing work focuses on dataset creation and prompt engineering. While engineering innovations such as the application of contrastive learning or MoE represent important technical contributions in our work, these methods have not previously been popular applied to cultural awareness tasks, and thus are not the main focus of prior research in this field. Including extensive discussions of these techniques in the related work section may introduce unrelated details from other tasks, which could confuse readers about the scope and context of our paper. Nevertheless, we appreciate the reviewer’s perspective and are willing to add brief introductions to key techniques such as contrastive learning and MoE in the appendix.
>
> **Q4:** I am confused with the concept of expert choice and routing mechanism...Could you explain more....
>
> **A4:** We are very happy to clarify these points in detail. First, we would like to clarify that we have systematically introduced the construction of the input in lines 96–147, including mathematical formulations, theoretical footnotes, and descriptive explanations. Lines 160–174 further formalize and explain the MoE workflow. All implementation details for every step of the methodology are provided in Appendix C.
>
> Here, we have provided additional explanation from the perspectives of theoretical priors, generation of the routable cultural representation, expert choice and routing mechanism including a simplified code snippet of the routing module, and why abstract features still enable dimensional specialization, which **will be provided to you immediately when the author-reviewer discussion period starts** (due to the space limitation in this rebuttal).
>
> **Q5:** ... do the favorable results stem from the underlying model itself? ... provide some experiment results based on different underlying models.
>
> **A5:** Thank you for asking this. We acknowledge that model performance often correlates with parameter scale, as indicated by scaling laws. However, numerous influential studies have shown that strong architectural design can enable smaller models to outperform those with larger parameter counts [3]. In addition, on some tasks, simply increasing the number of parameters does not guarantee better results, as evidenced in prior literature [4].
>
> If one attributes the strong results in Table 1 only to the parameter size of Qwen3-32B, it is difficult to explain why CALM achieves even better results than much larger proprietary models such as GPT-4 (as shown in Table 3, with an improvement of over 5\%), GPT-3.5 (an improvement of 9.65\% in Supplementary Appendix C), and GPT-4o (more than 20\% improvement in Table 4). These OpenAI models have substantially more parameters than Qwen3-32B, and the GPT-4 series in particular is widely considered among the strongest language models available. Even the official results reported by the Qwen3 team are only comparable to those of GPT-4. In our experiments, however, CALM built on Qwen3-32B is able to outperform GPT-4o by a wide margin. These empirical results support our claim that CALM’s improvements stem from its model design rather than parameter size. Furthermore, Table 4 in our paper includes models such as DistilBERT, ALBERT-V2, and BERT, as reported by the dataset authors. These models contain only a few hundred million parameters but, when fine-tuned on the target datasets, can outperform GPT-4o on certain tasks. This further demonstrates that thoughtful model design, rather than simply increasing the number of parameters, is essential for superior performance.
>
> To further address your question and to provide a more comprehensive evaluation of CALM, we have conducted additional experiments using various backbone models with different sizes and architectures, including but not limited to the baselines suggested by the reviewer.
>
> Due to the space limit in the rebuttal format, we were unable to add the further details (such as the experimental results and analyses for these backbone models) that we have prepared, but **will provide them to you immediately when the author-reviewer discussion period starts**.
>
> **Q6:** The use of footnotes is nonstandard.
>
> **A6:** As we explained in our response to Q1, placing foundational theoretical explanations in footnotes is a standard academic writing practice. Please refer to examples in [5, 6] which adopt this approach. This choice is not related to page limits or any other external factors. Using footnotes to highlight the key theoretical underpinnings of our design helps draw the reader’s attention to these important elements without interrupting the main flow of the text, and our use of footnotes fully complies with academic writing standards and NeurIPS formatting guidelines. We do not believe this should be considered a weakness of our submission (kindly note that we are experienced academic writers and have received prestigious best paper awards). Nevertheless, we respect the reviewer’s perspective and are willing to incorporate this content into the main text or the appendix in our revised submission, should the reviewer prefer that format.
>
> **Q7:** Irregular use of bold text in tables.
>
> **A7:** We acknowledge that in Table 4, the “Emissions” boldface should have been applied to ALBERT-V2, rather than CALM. This was an oversight on our part, and we will correct it in the revised version. For all other tables, the use of bold text is fully consistent: boldface always indicates the best performance, which is the standard convention in academic writing for computer science and artificial intelligence papers unless otherwise specified. Thank you.
>
> [1] D. Zhou, C. Brix, et al. Scalable neural network verification with branch-and-bound inferred cutting planes. NeurIPS, vol. 37, pp. 29324–29353, 2024.
>
> [2] Z. Shi, A. X. Yang, et al. Instruction tuning with loss over instructions. NeurIPS, vol. 37, pp. 69176–69205, 2024.
>
> [3] H. Touvron, T. Lavril, et al. LLaMA: Open and efficient foundation language models. 2023.
>
> [4] W. Fedus, B. Zoph, and N. Shazeer. Switch transformers: Scaling to trillion parameter models with simple and efficient sparsity. NeurIPS, vol. 34, pp. 8491–8505, 2021.
>
> [5] J. M. Swales and C. B. Feak. Academic Writing for Graduate Students: Essential Tasks and Skills. University of Michigan Press, 2004.
>
> [6] W. C. Booth, G. G. Colomb, and J. M. Williams. The Craft of Research. University of Chicago Press, 2016.

---

> ### Author Response · Authors · 2025-08-01
>
> **In accordance with our commitment in the “Rebuttal” block, we present here the remaining content for responses A4 and A5.**
>
> **Continuing from A4:**
>
> 1. Theoretical priors: Drawing on well-established research in cross-cultural communication, we decompose cultural differences into three complementary dimensions: contextuality (Hall's high/low-context theory), interpersonality (Brown \& Levinson's politeness theory), and normativity (Hofstede's and Schwartz's value theories). The specific meanings of these theories are detailed in footnotes 4, 5, and 6. This decomposition is not a mere set of labels, but rather corresponds to different linguistic levels: Contextuality relates to discourse-level information density and omission (see footnote 4); Interpersonality focuses on sentence-level politeness strategies (see footnote 5); Normativity involves lexical and syntactic preferences regarding values (footnote 6). These priors give each dimension clear linguistic phenomena as learning targets, which provides explicit guidance for the downstream expert networks.
>
> 2. Generation of the routable cultural representation $H_{\mathrm{align}}$ (see lines 95--147 and Appendix C for training details): MoE routing relies on a unified cultural representation $H_{\mathrm{align}}$, which is generated through a sequential process. First, the LLM encoding is decoupled into task semantics ($H_{\mathrm{task}}$), explicit cultural concepts ($H_{\mathrm{explicit}}$), and latent cultural signals ($H_{\mathrm{latent}}$). The explicit stream is optimized via masked reconstruction to learn idioms, honorifics, and other symbolic concepts, which are typically word- or phrase-level features (see Line 113 and Footnote 2). In contrast, the latent stream leverages style and value classification to capture cues such as tone, formality, and indirectness, usually at the sentence or discourse level (see Line 114 and Footnote 3). To promote cultural separation, separate contrastive losses are then applied to the explicit and latent streams, encouraging representations from the same culture to cluster together while pushing those from different cultures apart. Subsequently, Gumbel-Softmax clustering is used for each stream, and multi-head cross-attention aligns implicit (latent) and explicit clusters. This mechanism allows, for example, implicit "polite tone" clusters to be aligned with explicit "honorific" clusters. The resulting $H_{\mathrm{align}}$ encodes discourse structure, politeness strategies, and value cues, serving as the input to expert routing.
>
> 3. Expert choice and routing mechanism: For each dimension (contextuality, interpersonality, and normativity), we construct a pool of four experts, where each expert consists of two Transformer layers (hidden size 512, FFN size 2048) initialized according to the Qwen scheme, with weights drawn from $\mathcal{N}(0, 0.02)$ and biases set to zero to ensure stability. We adopt an “expert choice” principle in which, within each dimension, every expert scans the entire batch and computes an affinity score for each input. Each expert then selects the top-$K$ inputs it is most suited to process, and only these inputs are routed through the corresponding expert. The gating scores are softmax-normalized across the selected inputs for each expert, ensuring that activation remains sparse and competitive. To encourage balanced utilization, a load-balancing regularization term is applied, so that all experts are engaged across the batch. The selected experts generate specialized representations for their chosen inputs, which are weighted and aggregated. Outputs from all three dimensions are concatenated, passed through an MLP, and then combined residually with explicit and latent representations to form the unified cultural identity $H_{\mathrm{self}}$.
>
> 4. Why abstract features still enable dimensional specialization: Abstract features continue to support dimensional specialization through several complementary mechanisms. First, task-driven gradients from downstream objectives such as value recognition, politeness classification, and taboo detection provide distinct learning signals for each expert pool, guiding the model to focus on patterns specific to each dimension. Additionally, contrastive learning and cross-attention constraints maintain clear cultural cluster structures and ensure coherent alignment of implicit and explicit features within $H_{\mathrm{align}}$. Sparse activation and load balancing are also applied to each dimension by enforcing sparse dispatch loss and promoting even usage across experts, which prevents the model from collapsing to a single dominant expert.

---

> ### Author Response · Authors · 2025-08-01
>
> **Continuing from A4:**
>
> For illustration, we provide a simplified code snippet of the routing module:
>
> ```python
> import torch
> import torch.nn as nn
>
> class ExpertRouterEC(nn.Module):
>     def __init__(self, embed_dim, num_experts=4, top_k=2):
>         super().__init__()
>         self.num_experts = num_experts
>         self.top_k = top_k  # number of inputs each expert selects
>         self.gate = nn.Linear(embed_dim, num_experts, bias=True)
>         self.experts = nn.ModuleList([
>             nn.TransformerEncoderLayer(
>                 d_model=embed_dim,
>                 nhead=8,
>                 dim_feedforward=2048,
>                 batch_first=True
>             ) for _ in range(num_experts)
>         ])
>         nn.init.normal_(self.gate.weight, mean=0.0, std=0.02)
>         nn.init.zeros_(self.gate.bias)
>
>     def forward(self, H_align):
>         # H_align: (batch, seq_len, embed_dim)
>         batch_size = H_align.size(0)
>         device = H_align.device
>         h_bar = H_align.mean(dim=1)  # (batch, embed_dim)
>         scores = self.gate(h_bar)    # (batch, num_experts)
>         outputs = torch.zeros_like(H_align)
>         input_selection_count = torch.zeros(batch_size, self.num_experts, device=device)
>
>         for expert_id, expert in enumerate(self.experts):
>             # For this expert, find top-k input indices it wants to process
>             expert_scores = scores[:, expert_id]           # (batch,)
>             topk_val, topk_idx = torch.topk(expert_scores, self.top_k, dim=0)
>             # Softmax over the selected top-k scores
>             weights = torch.softmax(topk_val, dim=0)       # (top_k,)
>             # Gather the corresponding input representations
>             selected_inputs = H_align[topk_idx]            # (top_k, seq_len, embed_dim)
>             # Expert processes these inputs
>             expert_outputs = expert(selected_inputs)       # (top_k, seq_len, embed_dim)
>             # Aggregate weighted outputs back to the batch locations
>             for i, idx in enumerate(topk_idx):
>                 outputs[idx] += weights[i] * expert_outputs[i]
>                 input_selection_count[idx, expert_id] += 1  # For load balancing
>
>         # For load balancing, average number of times each expert selects any input
>         expert_selection_freq = input_selection_count.sum(dim=0) / batch_size  # (num_experts,)
>         load_balance_loss = (expert_selection_freq * self.num_experts).log().mean()
>
>         return outputs, load_balance_loss
>
> # Instantiate routers for each dimension
> router_ctx  = ExpertRouterEC(embed_dim=512)
> router_int  = ExpertRouterEC(embed_dim=512)
> router_norm = ExpertRouterEC(embed_dim=512)
>
> # Forward pass
> H_ctx,  lb_ctx  = router_ctx(H_align)   # Contextuality
> H_int,  lb_int  = router_int(H_align)   # Interpersonality
> H_norm, lb_norm = router_norm(H_align)  # Normativity
>
> # Aggregate the three losses
> total_load_balance_loss = lb_ctx + lb_int + lb_norm
>
> # Concatenate the three dimension-specific outputs
> H_self = torch.cat([H_ctx, H_int, H_norm], dim=-1)
>
> # Further processing follows as in the paper
>
> ```
>
> We hope this clarifies your concerns. If you have any further specific questions, we welcome continued discussion! Additionally, as we have stated in our response to Checklist Question 5 (lines 742–743), we are committed to open-sourcing the full model and code upon acceptance of the paper, so that readers can directly explore the internal mechanisms for a more intuitive understanding.

---

> ### Author Response · Authors · 2025-08-01
>
> **Continuing from A5:**
>
> | Model                    | Resource      | Precision | Recall | F1   |
> |--------------------------|--------------|-----------|--------|------|
> | **Llama-3.1-8B-Instruct**| All          | 90.5      | 68.9   | 78.1 |
> |                          | High         | 93.4      | 74.2   | 82.6 |
> |                          | Mid          | 89.2      | 71.7   | 79.3 |
> |                          | Low          | 86.1      | 64.4   | 76.1 |
> | **gemma-3-12b-it**       | All          | 87.0      | 70.3   | 77.6 |
> |                          | High         | 90.7      | 72.5   | 80.4 |
> |                          | Mid          | 85.6      | 74.2   | 79.4 |
> |                          | Low          | 83.9      | 67.2   | 76.2 |
> | **gemma-3-27b-it**       | All          | 92.2      | 76.4   | 84.6 |
> |                          | High         | 95.1      | 79.1   | 86.7 |
> |                          | Mid          | 91.0      | 77.8   | 84.7 |
> |                          | Low          | 90.3      | 74.6   | 81.8 |
> | **Qwen3-8B**             | All          | 89.7      | 67.8   | 76.8 |
> |                          | High         | 92.5      | 73.1   | 81.5 |
> |                          | Mid          | 88.3      | 69.8   | 77.9 |
> |                          | Low          | 86.9      | 62.1   | 72.7 |
> | **Qwen3-14B**            | All          | 93.3      | 71.9   | 81.2 |
> |                          | High         | 96.2      | 76.6   | 85.1 |
> |                          | Mid          | 91.9      | 74.1   | 82.0 |
> |                          | Low          | 90.8      | 67.5   | 76.5 |
>
> Our additional experimental results show that even when CALM is built upon backbone models with similar parameter scales, it still significantly outperforms the baseline models. Our key message is that a robust and well-designed framework should demonstrate generalizability and transferability. CALM can be combined with any large language model, such as Qwen, Llama, or Gemma, and at any parameter scale, including 8B, 14B, 32B, or even beyond. The strong and robust performance of CALM is attributable to its novel mechanisms, which provide stability and generalization independent of the model family or size. Under our approach, both large and small models benefit from our framework, advancing the field beyond simple scaling strategies. This is one of the most important contributions of our work. We will include these additional results with different backbones in the appendix of our revised version to provide greater clarity and transparency for readers.

---

> ### Comment · Reviewer_QL7m · 2025-08-03
>
> Thank you for your rebuttal. However, a key concern remains for me regarding the abstract cognitive space, which is the cornerstone of your framework. Why do you believe this abstract cognitive space is sufficient to handle the complexity of culture, and how is this achieved? The complexity involved in capturing explicit and implicit signals within cognitive science is vastly greater than what your framework addresses. The core weakness persists; neither your paper nor your clarifications justify your approach's validity. This gap makes me doubt whether your method can work as claimed, which is a significant barrier to my acceptance of the framework.

---

> > ### Author Response · Authors · 2025-08-04
> >
> > Thanks for your response. Regarding your core concern about the sufficiency and validity of our "Abstract Cognitive Space" (ACS) in modelling cultural complexity, we would like to address this from both conceptual and empirical perspectives.
> >
> > First, we wish to clarify that we do not claim our framework can handle all aspects of cultural complexity. In fact, we have explicitly acknowledged in our paper that no abstraction can fully encode every dimension of culture. As discussed in lines 325–332, "We recognize that culture is not a static entity, but a dynamic and multifaceted system shaped by history, interaction, and context. We also acknowledge that no existing approach can fully capture the vivid, evolving, and layered nature of human cultural identity." In Appendix A, we made a similar statement, noting that "Second, predefined cultural dimensions, despite dynamic selection, may oversimplify cultural diversity..." According to our literature review, including the recent surveys we cited ([50], [51]), there is currently **no model** capable of completely covering the full spectrum of culture. This is a fundamental challenge for the field and not one that we claim to have solved. Instead, we aim to alleviate this limitation as much as possible, and we believe our efforts represent a significant advance over prior models in several key respects:
> >
> > **1. Theoretical motivation. (Why Abstract Cognitive Space?)**
> >
> > Culture is indeed highly complex, involving explicit symbols, implicit norms, values, communicative strategies, and contextual practices. Our approach is inspired by leading research in sociocultural and cognitive linguistics, as cited in our paper, which consistently finds that meaningful cultural modelling must at least disentangle three levels: semantic intent (the propositional content), explicit cultural concepts (overt symbolic forms, such as idioms, honorifics, or role markers), and latent cultural signals (pragmatic, implicit, and discourse-level cues such as tone, indirectness, and stance). This tripartite decomposition is grounded in theories by Vygotsky, Halliday, Gumperz, and others, and reflects the way humans process and infer cultural information (see footnotes 1, 2, 3).
> >
> > **2. How is cultural complexity represented? (How is ACS constructed?)**
> >
> > Operationally, the ACS in CALM is not a simple vector but a structured, multi-channel representation. $H_{\mathrm{task}}$ encodes domain and task semantics, capturing goal-directed meaning. $H_{\mathrm{explicit}}$ extracts high-level explicit cultural concepts, focusing on socially salient idioms, honorifics, and role nouns. $H_{\mathrm{latent}}$ captures implicit, sentence- and discourse-level stylistic features, analysing cues such as formality, tone, and indirectness. This is not mere "feature engineering," but a learned disentanglement over large-scale data, further refined through joint training and contrastive learning to ensure that each stream captures unique and coherent aspects of cultural variation (see Appendix C for technical details).
> >
> > At the theoretical level, explicit cultural concepts and latent cultural signals together capture both the breadth and depth of cultural understanding. Explicit cultural concepts mainly refer to symbols, norms, institutions, and overt pragmatic rules, such as honorifics, role markers, and idiomatic expressions. These are typically formal rules, openly taught within a culture and readily abstracted through visible symbols. Examples include role titles like "Doctor" or "Madam", institutional expressions such as Japanese honorifics, and group identity terms, all of which reflect cultural dimensions such as hierarchy, politeness, and collective identity. In contrast, latent cultural signals include pragmatic and communicative features that are less easily defined but vital for cultural cognition. These comprise tone, indirectness, formality, ambiguity, and emotional nuance, serving as contextualization cues. Such cues often appear as implicit rules and habitual practices in real interactions. For instance, indirect speech is prevalent in high-context cultures, while humour, wordplay, and subtle implications reveal underlying cognitive and communicative styles. Explicit concepts and latent signals thus map onto two main cultural dimensions: one encompassing symbols, norms, and institutionalized behaviour, and the other covering pragmatics, style, cognition, and interaction. Theoretically, these dimensions are complementary. As emphasized in sociolinguistic and cross-cultural pragmatics research cited in our paper, cultural differences manifest not only in explicit symbols but also in deeper communicative styles and cognitive patterns. Both aspects are essential for a comprehensive model of culture.

---

> > ### Author Response · Authors · 2025-08-04
> >
> > **3. Task-oriented cultural features.**
> >
> > Our study focuses on two main categories of tasks, as stated in lines 208 to 210 of the manuscript. The first is knowledge-oriented tasks, which evaluate commonsense and values within cultural contexts. The second is toxicity-sensitive tasks, which assess the detection of culturally harmful content such as hate speech and social bias. The complementarity between explicit cultural concepts and latent cultural signals forms the foundation of our model design for both categories of tasks.
> >
> > In knowledge-oriented tasks, such as cultural commonsense reasoning and value identification, explicit cultural concepts enable the model to directly capture clear and codified knowledge. Examples include ceremonial language, social roles, and terminology related to festivals or institutions, all of which serve as explicit evidence for commonsense judgments and value assignments. For instance, determining whether a festival is unique to a particular culture or whether a title reflects social hierarchy depends on explicit semantic markers. Latent cultural signals, on the other hand, help the model capture stylistic and interactional features that cannot be easily formalized but are essential for commonsense reasoning in context. Examples of these signals include indirectness such as euphemisms and hedging, communicative styles that reflect power distance, and subtle cues indicating group identity. These implicit signals often determine whether a statement is regarded as commonsense or as a shared value within a culture.
> >
> > For toxicity-sensitive tasks, such as hate speech and bias detection, much of the harm in toxic or biased expressions is present in the use of sensitive symbols or stereotypical keywords. These belong to the domain of explicit cultural concepts, such as targeted slurs and discriminatory terms. The model must be able to detect and interpret these high-risk words and their contexts accurately. At the same time, many covert forms of toxicity and social bias are not expressed directly. Instead, they are embedded in tone, insinuation, sarcasm, or other latent cultural signals. Examples include sarcasm, passive aggression, or stereotypes disguised as humour, all of which are especially common in low-resource or high-context cultures. If a model only encodes explicit features, it is likely to miss these subtler forms of toxicity and thus fail to provide sufficient protection for minority groups.
> >
> > By combining explicit and implicit channels, our multi-channel modelling approach is able to capture both direct knowledge and symbolic cues, as well as the nuanced cultural risks that arise from style and context. This enables the model to achieve robust performance in both types of cultural tasks, ensuring comprehensive and sensitive cultural understanding.
> >
> > **4. Is this sufficient for cultural complexity?**
> >
> > It is important to clarify that addressing cultural complexity is not the responsibility of the ACS (Abstract Cognitive Space) alone, but rather the result of the entire CALM framework working together. Within CALM, each component operates in coordination with the others, and the ACS should not be evaluated in isolation. The ACS provides a minimal, theory-driven foundation that separates explicit and implicit cultural information from task semantics, supporting modular reasoning and ensuring alignment with established models in linguistics and artificial intelligence. The two cultural streams are explicitly distinguished and organized through contrastive learning and type-specific objectives, which increases both the diversity and the discriminative power of the resulting cultural representations. In the subsequent stages of the framework, the ACS serves as the input to the identity alignment pool, where explicit and implicit channels are aligned by cross-attention and further processed through a dimension-specific mixture-of-experts mechanism that encompasses contextuality, interpersonality, and normativity. This approach allows the model to specialize and generalize across various communicative dimensions. The ACS does not function as a static lookup table, but instead as a dynamic state that is recalibrated through a reflective reasoning loop. This process enables the model to identify and correct cultural misalignments as they arise in context. Through these integrated mechanisms, CALM is constructed as a holistic system capable of handling cultural complexity, rather than as a set of isolated or independent modules.

---

> > ### Author Response · Authors · 2025-08-04
> >
> > **5. Empirical validation. (Does this work in practice?)**
> >
> > Our approach is validated not only by strong generalization performance across four benchmark datasets and extensive experiments in the supplementary material, but also by comprehensive ablation studies. When either the latent or explicit stream is removed, there is a substantial drop in F1 score, specifically 1.3% and 2.0%, respectively. The greatest performance decrease is observed in low-resource and culturally variable environments. These results provide direct evidence that both streams are necessary to capture complementary aspects of culture. When contrastive learning is removed, the resulting clusters are less coherent and cultural confusion increases, which demonstrates that both streams are structurally necessary and not redundant. In addition, removing downstream modules of ACS, such as the identity alignment pool, the contrastive window, or the reflective reasoning mechanism, leads to a significant degradation of CALM's performance. This finding underscores the importance of integrating disentangled cultural signals for effective reasoning. Altogether, these results confirm our central position that CALM must be understood and evaluated as a complete system for handling cultural complexity, rather than attributing this capability to the ACS component alone.
> >
> > **6. Additional evidence for robustness and interpretability.**
> >
> > Our experiments further demonstrate robustness from multiple angles. In terms of interpretability and consistency, token-level attribution analysis (Appendix F and the supplementary material) shows that CALM produces method-agnostic and stable internal explanations. The high consistency between explanation paradigms (SHAP and LIME) indicates that its cultural signals are not arbitrary, but semantically meaningful and aligned with expert annotation. These findings are statistically significant. Supplementary Appendix B further provides visual comparisons (Figures 2 and 3) showing that, regardless of whether tokens are ranked by SHAP or LIME, the high-scoring tokens identified by CALM (such as "terrorist", "criminals", and "sexist") are not only highly consistent across methods, but are also the core semantic triggers for stereotype categories in the dataset. This demonstrates that CALM's abstract cognitive space captures and distinguishes culturally sensitive expressions that are semantically relevant, achieving method-invariant, semantically aligned interpretability and consistency. These analyses offer direct evidence for the semantic soundness and transparency of the model's internal reasoning structure, and validate the effectiveness and robustness of CALM's cultural representation. Additionally, in terms of instruction adherence (Supplementary Appendix D), CALM achieves zero out-of-choice errors, indicating that the cultural reasoning process embedded in ACS does not interfere with the correctness of task execution.
> >
> > We hope this further addresses your concern. We also hope this could convince you to accept our work. If you have any further question, please do let us know, and we will be more than happy to provide further clarification. Thank you.

---

> > > ### Comment · Reviewer_QL7m · 2025-08-05
> > >
> > > Thanks for your detailed feedback. I will consider them in the final evaluation.

---

> > > > ### Author Response · Authors · 2025-08-05
> > > >
> > > > We are glad to hear that your concerns have been addressed! We kindly ask you to consider revising your original score, as we believe it may not reflect the quality of our paper. If you have any remaining questions or uncertainties, we would be more than happy to provide further clarification during this Author-Reviewer discussion period, which has now been extended by NeurIPS.

---

### Official Review · Reviewer_z9ZR · 2025-06-30

**Clarity:** 3
**Significance:** 3
**Originality:** 4
**Rating:** 4
**Confidence:** 4

**Summary:**

The paper propose a novel framework to endow language models with cultural self-awareness. The experimental results verify the effectiveness of the method.

**Questions:**

1. How to train the model? What is the training data?
2. As shown in Table 1, the performance of your model is worse than ChatGPT on all cultures. The results is not good enough.
3. Regarding "ChatGPT", what is the version of "ChatGPT" in the paper?
4. No comparison with other cultural models, such as CulturalBank, CultureLLM. You can also compare with some culturally specific model, such as SeaLLM.

**Ethical Concerns:**

["NO or VERY MINOR ethics concerns only"]

**Limitations:**

Yes

**Quality:**

3

**Strengths And Weaknesses:**

Strength:
1. The paper propose a novel framework to endow language models with cultural self-awareness.
2. It introduced an interesting idea that teach model the deep representations of cultures rather than shallow representations.
3. The experiments setting is solid.

Weakness:
1. How to train the model? What is the training data?
2. As shown in Table 1, the performance of your model is worse than ChatGPT on all cultures. The results is not good enough.
3. Regarding "ChatGPT", what is the version of "ChatGPT" in the paper?
4. No comparison with other cultural models, such as CulturalBank, CultureLLM. You can also compare with some culturally specific model, such as SeaLLM.

---

> ### Author Rebuttal · Authors · 2025-07-30
>
> First of all, we sincerely appreciate your valuable feedback and thoughtful comments.
>
> **Q1:** How to train the model? What is the training data?
>
> **A1:** Please refer to lines 208–225 for a detailed introduction of the four datasets used in our work; references [52], [53], [54], and [55] cite the original dataset papers. For training procedures, please see Appendix C (Implementation Details), which provides comprehensive information on module implementation, inference strategies, hyperparameter settings, experimental environment, and more. If you are interested in resource consumption, Figure 3 summarizes the training time, number of epochs, and sample statistics. Additionally, in our response to reviewer MxFb Q4, we conducted further experiments to report VRAM usage, inference costs, and model parameters. If you have any specific concerns about the data or training process, we would be happy to address them in detail. We also include the pseudocode (Appendix B) and are committed to releasing the complete training scripts to ensure full reproducibility.
>
> **Q2:** ...the performance of your model is worse than ChatGPT on all cultures. The results is not good enough.
>
> **A2:** Thank you for your question. We would like to politely highlight that this might be a misunderstanding. Please see our detailed clarification below.
>
> We acknowledge that CALM is slightly outperformed by ChatGPT on the CultureAtlas dataset (Table 1). We did not avoid this fact, but reported it honestly and transparently (see lines 261–262): “Despite not relying on additional supervision, CALM approaches the performance of the proprietary model ChatGPT.” For all baseline results in Table 1, we did not re-run or reproduce the ChatGPT or other baseline results ourselves. Instead, we faithfully cited the scores reported by the original dataset paper, out of respect and appreciation for their contributions to the field. This means our comparison with ChatGPT is entirely based on published results, the authenticity and exact evaluation setup of which are determined by the original dataset creators.
>
> While model performance on CultureAtlas is an important metric, it is not the only consideration when evaluating cultural awareness. CALM demonstrates significant advantages over GPT-4 and GPT-4o on other important benchmarks: on the CREHate dataset, CALM outperforms GPT-4 by over 5\% (Table 3); on the EMGSD dataset, CALM exceeds GPT-4o by more than 20\% (Table 4). Furthermore, in the supplementary material (Appendix C, Table 1), CALM achieves the highest accuracy across all five countries on consensus posts in CREHate, clearly surpassing both GPT-4 and GPT-3.5. This demonstrates CALM’s superior ability to align with cross-cultural consensus judgments. In Appendix D (Table 2), CALM also achieves a good OOC rate, outperforming GPT-4 and GPT-3.5 in instruction adherence and output validity, consistently generating valid and well-formed answers for all inputs.
>
> We believe that a fair evaluation should not focus solely on a single aspect where the model is slightly weaker, but rather consider the comprehensive strengths that CALM demonstrates across multiple challenging, culturally sensitive tasks. CALM is an open model trained without proprietary data or instruction tuning, offering reproducibility and transparency, and yet already matches or surpasses the performance of much larger, proprietary models such as GPT-4 and GPT-4o. Even top-performing models from Alibaba, Google, Meta, and Anthropic do not always outperform previous models in every dimension, and GPT-series models are widely recognized as the strongest competitors in the field.
>
> Given the above clarification, we sincerely hope that the misunderstanding is cleared and you agree that CALM indeed has demonstrated such competitive performance on several important benchmarks, particularly considering its scale and open approach.
>
> **Q3:** ...what is the version of "ChatGPT" in the paper?
>
> **A3:** Thank you for raising this question. As noted above, the ChatGPT results reported in our tables are faithfully cited from the original dataset papers (CultureAtlas), without any modification on our part. The authors of the CultureAtlas dataset described their baseline as "a closed-source foundation model from OpenAI, trained with alignment data," but did not specify the exact version or API date of ChatGPT used in their experiments. Upon careful examination, we have identified its reference that appears to be incorrect. We have made efforts to contact the dataset authors for clarification, but have not received a response thus far. We have thoroughly verified all reported baselines in the dataset, and for all other open-source or versioned models included in our comparisons, we have carefully documented their versions to ensure transparency and fairness. We believe that faithfully reporting the original baseline data is essential to maintaining the integrity of the comparison. For full transparency and reproducibility, all results for CALM presented in this paper are obtained from our own experiments, with detailed implementation settings disclosed. For closed-source models like ChatGPT, we rely on the most authoritative and publicly documented sources currently available. We will continue our efforts to contact the dataset authors to clarify the exact ChatGPT version used, and will update our manuscript accordingly should we receive further information.
>
> **Q4:** No comparison with...CulturalBank, CultureLLM...also compare with...SeaLLM
>
> **A4:** We appreciate the reviewer’s familiarity with advanced methods in cultural awareness research, such as CulturalBank, CultureLLM, and SeaLLM. We have actually compared our approach with all three models. We have carefully reviewed and discussed the contributions and limitations of these prior works in our related work section. For instance, in the Introduction (lines 41–44), we introduce CulturalBank in our discussion of prior work, and in lines 47–51, we describe CultureLLM and discuss their limitations, as well as those of other cultural models like CulturePark. In the following paragraph (lines 52–56), we summarize the general limitations of existing methods in order to motivate our own framework. This constitutes a typical horizontal comparison in academic writing [1].
>
> Our CALM framework is designed to address a broader and more challenging set of objectives. As stated in lines 87–88, “To move beyond this dichotomy, CALM models culture as an internal, dynamic reasoning state.” We have thoroughly reviewed the CultureBank and CultureLLM datasets. CultureBank focuses primarily on knowledge-base fine-tuning and does not include country or language distinctions. CultureLLM’s experiments are based on WVS-derived data and cover only nine cultures. In contrast, CALM is evaluated across four comprehensive and diverse benchmarks: commonsense reasoning (CultureAtlas), value classification (UniVaR), hate speech detection (CREHate), and bias detection (EMGSD). These datasets are not only larger in scale but also more culturally diverse. For example, the WVS subset used in CultureLLM is only a part of the UniVaR dataset, and the integration of low-resource cultures in CultureLLM is less extensive than in CultureAtlas. CALM is developed specifically to overcome the limitations of previous cultural modeling approaches, such as their static, externalized, and label-driven nature, and the lack of dynamic reasoning and generalization ability. Our goal is to advance the state of the art rather than focus on single-task knowledge retrieval or narrow fine-tuning, which would not align with the scientific ambitions of our framework. Due to these fundamental differences in datasets, tasks, evaluation frameworks, and methodology, direct benchmarking between CALM and CultureBank or CultureLLM is not technically feasible and would not be scientifically meaningful. Such a comparison would fail to reflect the main strengths or intended contributions of our framework. By selecting the most challenging benchmarks and covering a much broader spectrum of cultures, our aim is to demonstrate both the breadth and forward-looking nature of our method.
>
> As for SeaLLM, our rationale is similar to that described above. it is a region-specific language model, as described in its original paper: “SeaLLMs, an innovative series of language models that specifically focuses on Southeast Asian (SEA) languages.” By contrast, CALM addresses a much wider range of languages, geographic regions, and cultures. Even so, our model has been indirectly compared with SeaLLM. In the UniVaR dataset study, SeaLLM was included as one of the multilingual LLMs used for value extraction and embedding analysis to test the robustness of the proposed UniVaR representation. Since our CALM framework outperforms UniVaR across all four value-oriented corpora and multilingual evaluations (see Table 2), CALM also surpasses the value extraction capabilities demonstrated by SeaLLM in this context. Moreover, as detailed in Appendix E of our supplementary materials (and as explained in our response to Reviewer MxFb Q5), we have provided extensive additional experimental evidence to validate the superior performance of CALM compared to UniVaR (and SeaLLM) across multiple aspects. These include language-level UMAP visualizations, the separation of value-related and non-value-related features, and robustness across different corpora. CALM demonstrates stronger generalization and value-oriented robustness in cross-lingual and cross-cultural domains.
>
> [1] John M. Swales and Christine B. Feak. Academic Writing for Graduate Students: Essential Tasks and Skills (3rd Edition). University of Michigan Press, Ann Arbor, MI, 2012.

---

> ### Author Response · Authors · 2025-08-04
>
> Thank you for your efforts. If you have any further questions or concerns for clarification, we would be very happy to do so during this author-reviewer discussion period.

---

> > ### Comment · Reviewer_z9ZR · 2025-08-05
> >
> > Thank you for your answers! I'd like to keep my score.

---

> > > ### Author Response · Authors · 2025-08-05
> > >
> > > We are glad to hear that your concerns have been addressed! If you have any further questions, we would be more than happy to provide further clarification during this Author-Reviewer discussion period, which has now been extended by NeurIPS. (It could be even better if you may consider slightly increasing your original score after the Author-Reviewer discussion period.) Many thanks again.

---

### Official Review · Reviewer_ABkN · 2025-07-03

**Clarity:** 3
**Significance:** 3
**Originality:** 3
**Rating:** 4
**Confidence:** 3

**Summary:**

The authors are creating a whole framework for addressing culture awareness in LLMs, they are depending on attentions and mixture of experts to enable the awareness of the LLMs and the framework can provide reasoning for the behavior.

**Questions:**

- Are you planning to opensource the code and provide simple approaches to apply?
- Why these exact llms for comparison?
- Would you illustrate aside from equations the idea behind IdentityAlignmentPool and would you justify the usage of contrastive window as I did not totally get why it was important to use?
- How would you define high, medium and low.resource? this should have been defined in the work

**Ethical Concerns:**

["NO or VERY MINOR ethics concerns only"]

**Final Justification:**

The authors clarified many comments.

**Limitations:**

Some limitations are mentioned by the authors.

**Paper Formatting Concerns:**

No problems

**Quality:**

3

**Strengths And Weaknesses:**

Strength:
- Novel approach, the effort done to make the framework is apparent.
- Good evaluation and ablation studies to show the work

Weaknesses:
- Following the equations and the methodology was a bit tiring for me. (also I would need the other reviewers to check the credibility of all the equations and its writing as some equations were very confusing)
- The framework is a bit complicated to apply in reality, so I wonder how the authors would provide it in an easy way to apply.

---

> ### Author Rebuttal · Authors · 2025-07-30
>
> First of all, we sincerely appreciate your valuable feedback and thoughtful comments.
>
> **Q1:** ...check the credibility of all the equations and its writing as some equations were very confusing.
>
> **A1:** Thank you very much. We fully understand that some of the mathematical formulations, especially in the core technical section, may be challenging to follow. This difficulty may arise from several factors: the cross-module dependencies of certain equations, the abstraction required to model cultural concepts, the intrinsic complexity of the mathematics involved, and the novelty of our overall design. To address these challenges, we have made substantial efforts to present the equations in a modular and context-aware fashion. For each module or new concept introduced, we explain the design rationale and provide clear definitions for all mathematical symbols. We also include cross-references, such as “in Eqn. (5)” at line 160, to help readers easily locate previous formulations. These writing decisions are carefully considered to improve readability and ensure conceptual continuity.
>
> Regarding the credibility of the equations, we would like to assure the reviewers that all mathematical content has been carefully reviewed by several experts in mathematics on our team. That said, we also recognize that mathematical exposition is inherently challenging, especially in a framework like CALM that introduces new components and technical novelties. We greatly appreciate your feedback and will continue refining the presentation of our equations in future revisions. This may include adding step-by-step derivations, more intuitive explanations of variables, and additional diagrams to support understanding.
>
> If you have any specific points of confusion, we would be happy to clarify them in detail. Should we have the opportunity to attend NeurIPS in person, we would also welcome the chance to discuss these equations face-to-face at the conference.
>
> **Q2:** Are you planning to opensource the code and provide simple approaches to apply?
>
> **A2:** Yes, absolutely. As stated in our response to Checklist Question 5 (lines 742–743), "Upon publication, the code will be officially released, along with scripts to reproduce all the key results described in the supplementary materials." We fully recognize the importance of open-sourcing for scientific transparency and community progress, and we view it as a core commitment.
>
> Following the paper acceptance, we will release a complete codebase along with comprehensive documentation, modular APIs, and end-to-end instructions for both training and inference. Preconfigured scripts will also be provided to simplify setup. To support a wide range of use cases, we will provide a user-friendly toolkit and several plug-and-play examples that allow practitioners to selectively enable or disable specific modules (such as Identity Alignment or Reflective Reasoning). This design aims to make it easy to adapt or deploy CALM in diverse environments without requiring extensive technical intervention.
>
> The code will be released under the MIT License to maximize flexibility for modification and redistribution. Furthermore, we are committed to actively maintaining the repository. If users report issues, we will engage with them directly to ensure that the project remains accessible, functional, and impactful in practice.
>
> Considering that some users may have limited computational resources, we have also explored lighter-weight variants of the framework using smaller backbones, while still achieving competitive results. This is already reflected in our response to Reviewer QL7m and the supplementary experiments.
>
> **Q3:** Why these exact llms for comparison?
>
> **A3:** We selected these particular LLMs for three main reasons. First, they are the official baselines provided by the benchmark datasets we used. These models are often specifically fine-tuned or configured for strong performance on the respective datasets and represent the previous state of the art. It is standard practice in both academic writing and competitive evaluation to compare against the strongest prior models. Demonstrating improvements over such baselines lends greater credibility to the effectiveness of our proposed design. Second, as mentioned in lines 237–238, “Our selection remains consistent with previous frameworks.” Using the same comparison models as prior work ensures fairness and continuity. We also faithfully report the performance numbers documented in previous publications, rather than re-running or modifying those baselines ourselves. This decision reflects our respect for prior work and allows the community to clearly isolate and evaluate the gains brought by our approach. Third, the selected LLMs, including GPT-4, GPT-4o, Orca-2, Flan-T5, BERT, Vicuna, and LLaMA, collectively represent the major categories of modern LLMs. These include open-source models, instruction-tuned models, and proprietary systems. This coverage ensures that our evaluation is representative and rigorous, capturing a wide range of architectures and training paradigms. We believe this selection provides a fair, transparent, and meaningful assessment of CALM’s performance across different scenarios.
>
> **Q4:** Would you illustrate aside from equations the idea behind Identity Alignment Pool
>
> **A4:** We are very happy to provide further clarification. In Section 3.3 (Identity Alignment Pool), we have already described the conceptual motivation behind each step, including in lines 131–132, 142–143, 145–147, and 166–168. We follow this practice throughout the paper to help readers understand the reasoning behind our design decisions.
>
> The goal of the Identity Alignment Pool is to endow the language model with a unified, culturally grounded identity representation that reflects the hierarchical nature of human cultural understanding. This design is inspired by theories in sociocultural cognition and psycholinguistics (as cited in the paper), which argue that deep cultural understanding goes beyond identifying isolated cues. It involves the structured integration of explicit symbolic concepts and implicit pragmatic signals into a coherent internal state.
>
> Due to the space limit in the rebuttal format, we were unable to add the further detailed illustration for this point that we have prepared, but **will provide them to you immediately when the author-reviewer discussion period starts**.
>
> **Q5:** would you justify the usage of contrastive window?
>
> **A5:** Yes, we are very happy to clarify the motivation and justification behind the use of the contrastive window. We have already provided both theoretical and empirical support in the paper. For example, in line 110, we state the primary rationale as “to enhance the structural regularity and discriminability of the abstract cognitive space.” Lines 114–116 further elaborate that “contrastive learning encourages culturally coherent subspaces by maximizing intra-cultural similarity and minimizing inter-cultural overlap.” In the ablation study (Section 4.2, Figure 5), we show that removing the contrastive window leads to a significant drop in both cultural separability and classification accuracy. Lines 293–300 provide qualitative analyses of this effect, such as: “indicating that cultural cues... form non-uniform clusters in the embedding space... Without this module, culturally adjacent classes become more confusable,” and “effective cultural understanding requires... structurally integrating them to capture the communicative logic of diverse cultural contexts.” Due to space limitations, we were unable to elaborate further in the main text.
>
> Here, we have provided additional clarification on the underlying theoretical and practical motivations from the perspectives of theoretical foundation, neural representation, generalization benefit, and empirical support, which **will be provided to you immediately when the author-reviewer discussion period starts** (due to the space limitation in this rebuttal).
>
> **Q6:** How would you define high, medium and low resource? This should have been defined in the work
>
> **A6:** Thank you for your suggestion. We acknowledge that the definition of high-, medium-, and low-resource settings was not explicitly stated in our paper. Due to the page limit, our initial assumption was that these terms would be familiar to readers who have reviewed the original benchmark dataset, where the classification was introduced. In lines 260–261, we briefly refer to low-resource settings as “where linguistic coverage is sparse and cultural norms vary more drastically.” However, we agree that this may not be sufficiently clear or intuitive for readers without direct access to the original dataset documentation. Below, we provide the detailed classification, which is based on the definitions presented in the dataset's original publication: Cultural groups are categorized as "high-resource," "medium-resource," or "low-resource" according to two primary criteria: (1) the availability of linguistic and digital resources, and (2) the overall level of socioeconomic development associated with each group. Specifically, high-resource groups (e.g., the United States, China, France, Spain, Japan) are characterized by large amounts of training data, extensive population coverage, and advanced digital infrastructure. Medium-resource groups (e.g., Turkey, Egypt, Iran, Malaysia, Argentina) have moderate data availability and representation. Low-resource groups (e.g., Laos, Bhutan, the Democratic Republic of the Congo, Serbia) refer to cultural communities with very limited data coverage. These often correspond to underrepresented or marginalized populations, either due to economic constraints or the use of minority or under-documented languages. We will make sure to clearly include this definition in the revision of the paper to avoid ambiguity.

---

> > ### Comment · Reviewer_ABkN · 2025-08-06
> >
> > thank you for all the clarifications, I will increase the score of the clarity.

---

> > > ### Author Response · Authors · 2025-08-06
> > >
> > > Thank you very much once again for your professional review and your great support!

---

> ### Author Response · Authors · 2025-07-31
>
> **In accordance with our commitment in the “Rebuttal” block, we present here the remaining content for responses A4 and A5.**
>
> **Continuing from A4:**
>
> We begin by applying Gumbel-Softmax clustering to the cultural features extracted via contrastive learning. Both explicit cultural concepts and latent cultural signals are clustered separately using this differentiable method. The core idea here is that cultural knowledge should not be modeled as flat token-level facts, but rather as structured and clustered representations. In real-world settings, cultural elements such as honorifics or taboos, as well as pragmatic strategies like indirectness or power distance, tend to appear in identifiable but internally coherent groups. Contrastive learning encourages representations to organize naturally into clusters, but only through clustering can this structure be made explicit and accessible to downstream modules such as cross-modal interaction or expert routing. The differentiability of Gumbel-Softmax allows this clustering process to be trained end-to-end with the rest of the model, ensuring that it captures not only static groupings but also dynamically adapts to the input sample and task.
>
> Next, we apply multi-head cross-attention between the explicit and latent cultural clusters to generate aligned representations. For example, this allows the model to connect explicit entities such as family roles or religious groups with latent cues like deferential tone or indirect speech. The motivation here is that cultural understanding fundamentally relies on the integration of symbolic and pragmatic dimensions. Many cultural norms can only be interpreted correctly when both levels are considered together. Token- or phrase-level features alone cannot capture such higher-order cultural logic. Cluster-level cross-attention enables the model to move beyond isolated fragments and learn more abstract co-occurrence patterns, such as "religious entity + imperative tone." The use of multiple attention heads allows the model to capture many-to-many mappings between symbolic and pragmatic dimensions, which better reflects the richness of cultural phenomena in the real world.
>
> To account for the multidimensional nature of cultural variation, we organize the aligned features along three theory-driven cultural dimensions: contextuality, interpersonality, and normativity. For each dimension, we allocate a dedicated expert pool. In our routing mechanism, each expert actively scans the entire input batch, computes an affinity score with each input, and then selects only the top-k inputs it is most suited to process. This approach is grounded in the assumption that cultural variation follows interpretable axes, with distinct phenomena dominated by specific dimensions. For example, collectivist tendencies in East Asian contexts are often associated with high contextuality, while cross-cultural differences in power distance are captured within the broader dimension of normativity. Assigning each dimension to a separate expert pool reduces interference and promotes specialization. This design is inspired by the psychological principle of division of cognitive labor [1], which posits that both in the brain and in society, specialized units lead to more efficient and robust reasoning. Routing by expert choice within each cultural dimension outperforms conventional token-level routing by allowing experts to focus on culturally relevant patterns, reducing competition, and encouraging the emergence of meaningful expert clusters. By enabling each expert to select only the top-k inputs, rather than processing all inputs, we further reinforce specialization and minimize redundancy.
>
> Finally, we use a multilayer perceptron (MLP) to integrate the outputs from all cultural dimensions, and apply residual connections to preserve the important cultural signals from the original representation. This design allows the model to absorb the high-level reasoning provided by the dimension-specific experts, while retaining foundational cultural features. The residual mechanism, inspired by architectures such as ResNet and Transformer, helps prevent the loss of useful information during deep fusion. Compared to simple averaging or weighted summation, the use of concatenation followed by an MLP enables nonlinear integration of heterogeneous expert information, which supports more complex cultural interactions. The resulting identity representation thus captures both the breadth and depth of cultural knowledge, providing a strong foundation for reflective and culturally sensitive reasoning across the CALM framework.
>
> [1] P. Kitcher. The Division of Cognitive Labor. The Journal of Philosophy, Vol. 87, No. 1, pp. 5–22, 1990.

---

> ### Author Response · Authors · 2025-07-31
>
> **Continuing from A5:**
>
> 1. Theoretical foundation: Structured and separable cultural representations. Cultural features are not isolated points, but tend to form dense semantic groupings in the embedding space. Linguistic phenomena such as self-deprecating expressions or kinship-related terms exhibit high local coherence. This distributional structure aligns with categorization theory in psychology [2], which suggests that human cognition naturally organizes complex concepts into structured groups. Contrastive learning is a principled mechanism for learning such group structures. By pulling together representations of the same cultural category and pushing apart different ones, it induces clear cultural boundaries in high-dimensional space. Without this, models struggle to identify intra-group homogeneity and inter-group heterogeneity in cultural features.
>
> 2. Neural representation: Correcting cultural dilution in pretraining. Large language models trained with self-supervised objectives often optimize for average predictive likelihood over the next token or span. This emphasis on statistical regularities can dilute fine-grained cultural distinctions, since models tend to prioritize universal linguistic patterns rather than culturally specific cues. The contrastive window explicitly clusters both explicit and latent cultural features, enforcing semantic separation in the representation space. This structural encoding enables downstream modules to “retrieve by culture” and mitigates the loss of cultural variation during generalization.
>
> 3. Generalization benefit: A structural prerequisite for downstream modules. Modules such as Gumbel-Softmax clustering, cross-attention, and MoE-based expert routing rely on structured and disentangled input representations to function effectively. Without the contrastive window, cultural features may appear entangled and indistinguishable, undermining the ability of downstream components to make selective decisions. The contrastive window pre-aggregates each cultural cluster, giving experts a form of spatial awareness to identify and specialize in relevant cultural dimensions. This directly improves routing selectivity and specialization, which are critical for cultural reasoning.
>
> 4. Empirical support: Evidence from prior state-of-the-art contrastive learning methods. Several leading studies have demonstrated the unique effectiveness of contrastive learning in discovering latent subgroup structures in complex environments. Notable examples include SimCLR [3], MoCo [4], and CLIP [5], all of which show that contrastive objectives facilitate the emergence of meaningful structures under challenging settings. These insights strongly support our contrastive window design within the CALM framework.
>
> [2] E. Rosch and B. Lloyd. Cognition and Categorization. Taylor \& Francis, 2024.
>
> [3] T. Chen, S. Kornblith, M. Norouzi, and G. Hinton. A Simple Framework for Contrastive Learning of Visual Representations. ICML, pp. 1597–1607, 2020.
>
> [4] K. He, H. Fan, Y. Wu, S. Xie, and R. Girshick. Momentum Contrast for Unsupervised Visual Representation Learning. CVPR, pp. 9729–9738, 2020.
>
> [5] S. Radford, J. Wook Kim, et al. Learning Transferable Visual Models From Natural Language Supervision. ICML, pp. 8748–8763, 2021.

---

> ### Author Response · Authors · 2025-08-04
>
> Thank you for your efforts. If you have any further questions or concerns for clarification, we would be very happy to do so during this author-reviewer discussion period.

---

### Official Review · Reviewer_MxFb · 2025-07-03

**Clarity:** 3
**Significance:** 2
**Originality:** 3
**Rating:** 4
**Confidence:** 4

**Summary:**

This paper introduced Culturally Self-Aware Language Models (CALM) to integrate cultural knowledge into language models. They first disentangle task semantic from explicit cultural concepts and latent cultural signals. It then aligns these signals via contrastive learning, cross-attention and mixture of experts with culturally grounded expert routing mechanism. Extensive experiments are performed across four benchmarks, covering commonsense recognition, value identification, hate-speech detection, and social-bias detection, to show the effectiveness of the proposed method.

**Questions:**

See the above weaknesses section for questions.

**Ethical Concerns:**

["NO or VERY MINOR ethics concerns only"]

**Final Justification:**

The authors addressed my concerns, by providing consistent results across backbones.

**Limitations:**

Yes

**Paper Formatting Concerns:**

No formatting concerns.

**Quality:**

3

**Strengths And Weaknesses:**

Strengths:
- The proposed CALM outperforms strong open-source LLM baselines, and surpasses GPT-4 on the CREHate dataset across five countries.
- Extensive ablation study to show the impact of different components in Table 5.

Weaknesses:
- The comparison setup is not sufficient from Table1 to Table4. First the actual backbone detail is hidden in Appendix C line 545, which should be highlighted in the main text. As Qwen3-32B is used as the backbone, why not reporting results with a vanilla Qwen3-32B model in all the tables? This way we can see the impact of CALM on top of Qwen3-32B, instead of comparing Qwen3-32B+CALM with other models like (llama2, vicuna, flan-t5, opt, etc.). It could just be that Qwen3 is a more advanced model than some of the LLMs published long ago, which makes the results less convincing.
- CALM introduced additional complexity to the model architecture with additional encoders and experts, which increases memory cost, inference cost and parameters. Please also report results about the performance cost with CALM, to make it a good reference for practitioners.
- For multilingual evaluation tasks (UniVaR) in this paper, how does the proposed CALM perform across different languages? Additional per language results (in Appendix) could be useful.

---

> ### Author Rebuttal · Authors · 2025-07-30
>
> First of all, we sincerely appreciate your valuable feedback and thoughtful comments.
>
> **Q1:** backbone detail...should be highlighted in the main text.
>
> **A1:** We originally intended to include this in the main text, but due to space constraints, the complete implementation details (including the backbone specification) were moved to Appendix C. Line 225 of the main paper directs readers: "The implementation details are listed in Appendix." That said, we would like to emphasize a broader point: the backbone model is merely a tool for implementation, not the centerpiece of the contribution [1,2]. Our design is modular and agnostic to any particular foundation model. It can be Qwen, LLaMA, Gemma, DeepSeek, or others. As shown in our supplemental experiments (see response to reviewer QL7m), our architecture generalizes across multiple backbones. Selecting a specific model often comes down to practical preference, not scientific necessity. The focus of the paper is on our design rationale, technical contributions, and empirical findings, which are where the core novelty resides. Nonetheless, we respect the reviewer’s concern and are happy to prominently highlight the backbone detail (i.e., Qwen3-32B) in the main text of the revised version.
>
> **Q2:** ...why not reporting results with a vanilla Qwen3-32B model in all the tables?
>
> **A2:** We chose not to include the vanilla Qwen3-32B baseline in all performance tables because we consider it part of our ablation study rather than an independent comparison baseline. Readers interested in understanding CALM's improvements over Qwen3-32B can refer to the ablation results in Table 5, which show the progressive gains contributed by each module. Furthermore, introducing vanilla Qwen3-32B into the main result tables could create confusion. Readers might misinterpret it as a competing method, when it is in fact only the unmodified backbone. This could obscure the primary contributions of our paper, particularly the advances in design and methodology, and could disrupt the intended logical structure and narrative clarity, ultimately affecting the reader’s comprehension and experience.
>
> **Q3:** ...instead of comparing Qwen3-32B+CALM with other models... It could just be that Qwen3 is a more advanced model.
>
> **A3:** We explained our rationale for selecting these comparison models in lines 237–238: “Our selection remains consistent with previous frameworks.” The reason for this choice is to ensure a fair comparison. Therefore, all comparison models listed in Tables 1 to 4, such as LLaMA2, Vicuna, Flan-T5, and OPT as noted by the reviewer, are not arbitrarily chosen nor the result of a strategic trick. They are the official baselines provided by the benchmark datasets, and we faithfully report their results as originally documented, without re-running or modification. In many cases, these baselines are not raw pre-trained models; they have been fine-tuned or otherwise adapted to the datasets. Including an unadapted Qwen3-32B as a comparison would not be fair or meaningful. In this context, CALM is the appropriate point of comparison. It is our responsibility to compare with these established baselines in order to demonstrate the improvements introduced by CALM. This approach aligns with the standard expectations of competitive benchmarking and academic writing.
>
> Regarding the reviewer’s concern that CALM might only perform better because Qwen3 is more advanced than older models, we acknowledge that newer LLMs often have stronger capabilities. However, evaluating how a new model generalizes to culturally grounded tasks is a meaningful contribution in its own right. Moreover, we would like to reiterate the point made in our response to Q1: the backbone is not what drives our results. It could be any LLM, and we do not advocate for Qwen3 in particular. Our ablation study shows that CALM’s performance gains are progressive and interpretable. Without the CALM modules, the base model lacks the ability to perform strong cultural reasoning, regardless of whether it is built on a newer or older foundation. As shown in Appendix F, the performance gains are statistically significant and are driven by the design of our method, not by merely using a newer model. This is not a trick.
>
> **Q4:** ...increases memory cost, inference cost and parameters... Please also report results about the performance cost with CALM
>
> **A4:** Certainly. We have actually included the training time, number of epochs, and sample sizes in Figure 3, and carbondioxide emissions in Table 4. As requested by the reviewer, we have conducted additional experiments to calculate memory cost, inference cost, and parameter count, as shown in the table below:
>
> | Model Params (B) | Inference VRAM (GB) |
> |:----------------:|:------------------:|
> | 32.92            | >67                |
>
> | Dataset      | Inference FLOPs/sample (T) |
> |--------------|:-------------------------:|
> | UniVaR       | 1.29                      |
> | CultureAtlas | 2.59                      |
> | CREHate      | 0.65                      |
> | EMGSD        | 1.01                      |
>
> We are happy to include any additional metric the reviewer would like to see.
>
> **Q5:** For multilingual evaluation tasks (UniVaR)...how does the proposed CALM perform across different languages?
>
> **A5:** Appendix E of the supplementary material provides evidence that CALM outperforms the baselines across different cultural value frameworks and demonstrates its robustness to non-value factors such as translation artifacts and linguistic style. Put simply, language has little influence on CALM. To further verify this, we have conducted the following supplementary experiments and additional analyses:
>
>   1. UMAP Visualization: We apply UMAP to project high-dimensional value embeddings from 15 LLMs across 25 languages into a two-dimensional space. The resulting clusters exhibit strong cultural coherence. For instance, Chinese, Japanese, and Korean form a Confucian cluster; Arabic, Persian, Malay, and Indonesian group into an Islamic cluster; and English, French, German, and Spanish are grouped into a Western cluster. This structure closely mirrors the Inglehart–Welzel World Cultural Map [3], suggesting that the value representations of LLMs are deeply influenced by both language and the underlying cultural systems.
>
>   2. Comparative Evaluation: We compared CALM's performance on value-oriented QA, non-value QA, and translation source detection. CALM performs significantly better on the former tasks, indicating that it captures value semantics rather than superficial linguistic styles. This further validates its ability to effectively disentangle value-relevant information from confounding non-value signals. Some of the results are shown in the table below:
>
> |Model Name|WVS (k-NN)|WVS (Linear)|PVQ-RR (k-NN)|PVQ-RR (Linear)|GLOBE (k-NN)|GLOBE (Linear)|ValuePrism (k-NN)|ValuePrism (Linear)|
> |---|---|---|---|---|---|---|---|---|
> |BERT|1.15%|8.57%|2.99%|11.34%|1.88%|7.45%|1.11%|14.92%|
> |RoBERTa|1.36%|7.82%|2.83%|10.94%|1.95%|6.99%|1.39%|14.51%|
> |XLM-R|0.75%|7.12%|2.53%|8.85%|1.56%|6.23%|0.76%|12.38%|
> |MPNet|0.83%|4.36%|1.75%|4.49%|1.49%|2.86%|1.51%|8.47%|
> |LaBSE|2.44%|9.97%|3.69%|11.55%|3.61%|9.31%|4.08%|16.20%|
> |UniVaR|21.10%|19.14%|17.53%|16.34%|21.34%|18.66%|21.51%|20.55%|
> |**CALM**|**23.27%**|**21.33%**|**19.84%**|**18.71%**|**23.69%**|**20.89%**|**24.28%**|**22.93%**|
>
> |Model Name|text-only Acc@1|text-only Acc@5|paraphrase Acc@1|paraphrase Acc@5|
> |---|---|---|---|---|
> |BERT|17.22%|66.84%|26.97%|72.63%|
> |RoBERTa|15.20%|66.76%|19.98%|69.93%|
> |XLM-R|17.59%|67.37%|21.79%|70.40%|
> |MPNet|15.33%|65.85%|26.73%|72.13%|
> |LaBSE|14.66%|68.05%|23.95%|72.44%|
> |UniVaR|8.33%|58.73%|17.12%|63.16%|
> |**CALM**|**7.95%**|**56.80%**|**15.98%**|**62.17%**|
>
>   3. Language-Level UMAP: We visualize the embeddings of CALM and baseline models separately for each of the 25 languages. Each subplot corresponds to a single language, revealing the structure of value representations both within the same language and across different models.
>
> All of these findings are consistent across the WVS, PVQ-RR, GLOBE, and ValuePrism corpora. In other words, CALM exhibits stable performance across different languages, whereas the baseline models are more sensitive to cross-lingual variation. This is because CALM achieves stronger decorrelation and suppression of “non-value features” (such as translationese and source language) at the embedding level, which passively reduces its ability to encode source language information. These experiments highlight CALM’s ability to generalize value representations rather than language-specific features. Following the reviewer’s suggestion, we will include these visualizations, tables, and qualitative analyses in the appendix of the revised version.
>
> [1] Raffel, C., Shazeer, N., et al. (2020). Exploring the Limits of Transfer Learning with a Unified Text-to-Text Transformer. Journal of Machine Learning Research, 21(140), 1–67.
>
> [2] Liu, Z., Mao, H., et al. (2022). A ConvNet for the 2020s. CVPR, 11976–11986.
>
> [3] Inglehart, R. (2006). Mapping global values. Comparative Sociology, 5(2–3), 115–136.

---

> > ### Comment · Reviewer_MxFb · 2025-08-07
> >
> > Thank the authors for their response, as well as additional results across multiple backbones provided in the response to reviewer QL7m. I have read them, and don't have further questions. I will consider them in the final justification.

---

> > > ### Author Response · Authors · 2025-08-07
> > >
> > > Thank you very much for your time and careful consideration of our response. We are glad to hear that your concerns have been addressed, and we greatly appreciate your willingness to take our clarifications into account during the final evaluation. Should you have any further questions at any stage, we are always happy to provide more details and clarification.

---

> ### Author Response · Authors · 2025-08-04
>
> Thank you for your efforts. If you have any further questions or concerns for clarification, we would be very happy to do so during this author-reviewer discussion period.

---

### Note · Authors · 2025-08-11

We again sincerely thank all the reviewers, the ACs, and the committee for their careful consideration.

**1. Contributions and Significance.** All reviewers, in the "Strengths" section, agreed that CALM is novel/interesting, delivers strong performance, and presents clear experiments and ablation studies. CALM is grounded in well-recognized socio-cultural and linguistic theories and adopts a closed-loop architecture of four modules: cultural perception, structural induction, identity construction, and reflective correction. It originally and novelly models cultural awareness as a dynamic internal reasoning process, fundamentally different from prior approaches. On four challenging benchmarks and representative tasks, CALM consistently outperforms advanced LLMs and carefully fine-tuned models. The appendix and supplementary material provide further robustness and generalization validations in detail.

**2. Clarifications during the discussion.** Regarding backbone choice, we clarified that CALM is highly generalizable, independent of specific models or scales, and verified through additional experiments. For baselines, we faithfully used the models and results from the original papers, without employing any strategy to artificially emphasize the advantages of our own model, thereby ensuring fair and transparent evaluation. In methodology explanation, the high originality and interdisciplinary integration of CALM might make some concepts difficult to understand at first encounter. Apart from the detailed explanations provided in the paper, the rebuttal further elaborated on design motivation, theoretical foundations, training objectives, input features, inter-module information flow, and empirical validation. Other perceived weaknesses partly stemmed from reviewers' misinterpretations, which we addressed by pointing to exact relevant sections and conducting additional experiments. One reviewer considered the use of footnote in our writing a weakness, which became clear to the reviewer following our clarification. Overall, all the reviewers rated the *Clarity* of our paper good.

**3. Commitment to integration.** We will incorporate the supplementary material and all additional experiments from the rebuttal into the revised main text/appendices.

In sum, we actively engaged with the reviewers and resolved all concerns! By the end, **all reviewers confirmed that our clarifications addressed their concerns and raised no further questions**.

---

### Decision · Program_Chairs · 2025-09-17

**Decision:**

Accept (poster)

**Comment:**

This paper introduces a novel framework for incorporating cultural self-awareness into LLMs. The framework leverages contrastive learning, mixture-of-experts, and other mechanisms to integrate cultural knowledge, thereby enabling LLMs to handle diverse cultural contexts more effectively. The authors present robust experimental results, demonstrating that CALM outperforms various state-of-the-art models across several culturally sensitive tasks, such as commonsense reasoning, bias detection, and hate speech classification. The contributions are evaluated through extensive ablation studies, which highlight the significance of each component in the framework. While the novelty and technical soundness of CALM are widely acknowledged by reviewers, some concerns were raised. These included issues with clarity, especially regarding the complexity of the network, the model’s reliance on specific backbones, and comparisons with other cultural models. However, the authors addressed these concerns effectively in their rebuttal, clarifying the methodology, providing additional results, and improving the manuscript's presentation. Notably, despite some reservations about the framework's ability to fully capture the complexity of cultural phenomena, the majority of reviewers recognized CALM's potential as a significant advancement in the field.